# Epigenetic signals that direct cell type–specific interferon beta response in mouse cells

Markus Muckenhuber[1,2], Isabelle Seufert[1,2], Katharina Müller-Ott[1], Jan-Philipp Mallm[1,3], Lara C Klett[1,2], Caroline Knotz[1], Jana Hechler[1], Nick Kepper[1], Fabian Erdel[1], Karsten Rippe[1]

The antiviral response induced by type I interferon (IFN) via the JAK-STAT signaling cascade activates hundreds of IFN-stimulated genes (ISGs) across human and mouse tissues but varies between cell types. However, the links between the underlying epigenetic features and the ISG profile are not well understood. We mapped ISGs, binding sites of the STAT1 and STAT2 transcription factors, chromatin accessibility, and histone H3 lysine modification by acetylation (ac) and mono-/tri-methylation (me1, me3) in mouse embryonic stem cells and fibroblasts before and after IFN$\beta$ treatment. A large fraction of ISGs and STAT-binding sites was cell type specific with promoter binding of a STAT1/2 complex being a key driver of ISGs. Furthermore, STAT1/2 binding to putative enhancers induced ISGs as inferred from a chromatin co-accessibility analysis. STAT1/2 binding was dependent on the chromatin context and positively correlated with preexisting H3K4me1 and H3K27ac marks in an open chromatin state, whereas the presence of H3K27me3 had an inhibitory effect. Thus, chromatin features present before stimulation represent an additional regulatory layer for the cell type–specific antiviral response.

## Introduction

Type I interferon (IFN) cytokines like IFN$\alpha$ and IFN$\beta$ are expressed across almost all tissues in human and mouse as a first line of defense against viral infections (Hoffmann et al, 2015; Lazear et al, 2019; Sa Ribero et al, 2020; Stanifer et al, 2020). They activate hundreds of IFN-stimulated genes (ISGs) during innate immune response. Virus infection induces IFN$\beta$ in most cell types, which then can stimulate production of other type I IFNs (Hoffmann et al, 2015). However, ISG activation is not uniform but occurs in a cell type–specific manner (Lazear et al, 2019; Sa Ribero et al, 2020; Stanifer et al, 2020) and displays

striking changes during differentiation of human embryonic stem cells (Wu et al, 2018). Mouse embryonic stem cells (ESCs) do not express IFN themselves upon viral infection but respond to IFN and display an attenuated innate immune response as compared with differentiated murine cells (Whyatt et al, 1993; Gonzalez-Navajas et al, 2012; Wang et al, 2013; Wang et al, 2014; Guo et al, 2015; D'Angelo et al, 2016; Guo, 2017). Because the cell type–specific gene expression programs are dependent on the cell's epigenetic makeup, the associated chromatin features directly or indirectly affect ISG activation via the JAK-STAT signaling cascade. This pathway involves phosphorylation of STAT1 and STAT2 transcription factors that, together with IRF9, assemble into the IFN-stimulated gene factor 3 (ISGF3) complex (Stark & Darnell, 2012; Ivashkiv & Donlin, 2014; Chen et al, 2017; Villarino et al, 2017; Au-Yeung & Horvath, 2018; Hu et al, 2021). ISGF3 translocates into the nucleus, binds interferon-stimulated response elements (ISREs), and activates ISGs. In addition, IFN$\gamma$ activation sites (GAS) are bound predominantly by phosphorylated STAT1 homodimers and can also drive IFN-mediated gene induction. The STAT-binding sites are frequently located at promoters and regulatory sites such as enhancers (Vahedi et al, 2012; Ostuni et al, 2013; Begitt et al, 2014). Previously, it has been shown that chromatin remodeling complexes, histone acetyltransferases, and deacetylases can act as modulators for the JAK-STAT signaling cascade (Liu et al, 2002; Nusinzon & Horvath, 2003; Testoni et al, 2011; Chen et al, 2017; Villarino et al, 2017; Au-Yeung & Horvath, 2018). However, it is not well understood how cell type–specific epigenetic programs and chromatin features link STAT1 and STAT2 binding at cis-regulatory promoter and enhancer sequences to ISG induction.

Here, we dissected the cell type–specific IFN$\beta$ response by comparing mouse ESCs and MEFs. The comparison of ESCs to their differentiated counterparts is a well-established cellular system to distinguish DNA sequence versus epigenetically driven chromatin interactions of regulatory factors ([Teif et al, 2012; Teif et al, 2014; Thorn et al, 2022] and references therein). In the present study, we exploited it to assess the relation of ISG induction upon IFN$\beta$ treatment, binding of STAT1 and STAT2 and acetylation (ac) and

[1]Division of Chromatin Networks, German Cancer Research Center (DKFZ) and Bioquant, Heidelberg, Germany   [2]Faculty of Biosciences, Heidelberg University, Heidelberg, Germany   [3]Single Cell Open Lab, German Cancer Research Center (DKFZ), Heidelberg, Germany

Correspondence: karsten.rippe@dkfz.de
Markus Muckenhuber's present address is Disease Area Oncology, Novartis Institutes for BioMedical Research, Basel, Switzerland
Katharina Müller-Ott's present address is Illumina Centre, Granta Park, Cambridge, UK
Jana Hechler's present address is Technische Universität Nürnberg, Nürnberg, Germany
Nick Kepper's present address is Bioquant, Heidelberg University, Heidelberg, Germany
Fabian Erdel's present address is MCD, Centre de Biologie Intégrative (CBI), University of Toulouse, CNRS, Toulouse, France

mono- and tri-methylation (me1, me3) of histone H3 lysine residues (H3K4me1, H3K4me3, H3K9ac, H3K27ac, H3K9me3, and H3K27me3), and open chromatin mapped by the assay for transposase-accessible chromatin (ATAC). The resulting sets of common and cell type–specific ISGs were linked to the binding of a STAT1–STAT2 complex (STAT1/2) at promoters and enhancers in dependence of their chromatin state. Our analysis sheds light on the interplay of epigenetic signals, STAT1/2 binding at cis-regulatory elements, and the cell type–specific modulation of innate immune response.

# Results

### IFNβ induces anti-viral gene expression programs in all three cell types

ESCs, MEFs, and neural progenitor cells (NPCs) derived by in vitro ESC differentiation were obtained from a 129/Ola mouse strain, as described previously (Teif et al, 2012; Mallm et al, 2020) (Figs 1A and S1A). We selected 500 U/ml IFNβ and 1 and 6 h time points based on the strong induction of selected genes (IRF1, IRF3, IRF7, and ISG15) in ESCs, which is similar to conditions used in other studies (Burke et al, 2011; Schwerk et al, 2013; Wang et al, 2014; Bolivar et al, 2018; Platanitis et al, 2019). To characterize the genome-wide transcription response, we conducted an analysis by RNA sequencing (RNA-seq) (Table S1). Differential gene expression analysis identified up-regulated genes at 1 and 6 h of IFNβ stimulation, yielding a total of 191 ISGs in ESCs, 463 ISGs in MEFs, and 244 ISGs in NPCs over unstimulated controls (0 h time point) (Figs 1B and C and S1B and Tables S2 and S3). As expected, a GO-term analysis retrieved up-regulated genes related to anti-viral programs and innate immune responses in all three cell types (Fig S1C). By intersecting the three individual ISG sets, we obtained 143 common ISGs, whereas 33 (ESC), 17 (NPC), and 221 (MEF) ISGs were cell type specific (Figs 1C and S1B and Table S3). The ISGs

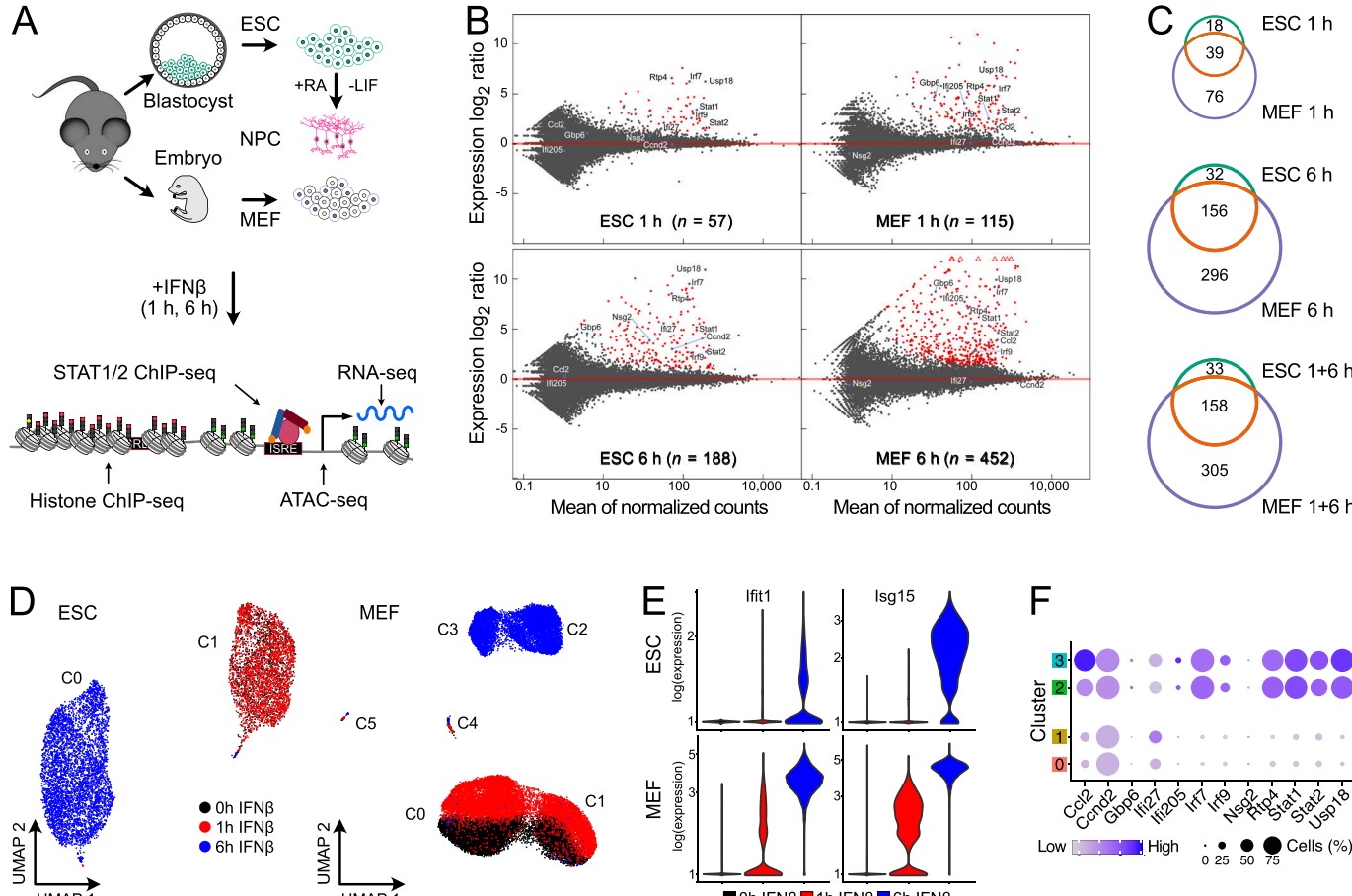

**Figure 1. ISG induction patterns in ESCs and MEFs.**
**(A)** ESCs, NPCs differentiated in vitro from them and MEFs from the same mouse strain were studied to reveal the relation between cell type–specific chromatin features and IFNβ response. **(B)** Gene expression changes after IFNβ treatment. Red dots represent significant differentially expressed genes at $P_{adj} < 0.05$ and fold change ≥1.5 as computed with DESeq2. Four biological replicates for ESCs and two for MEFs were acquired for RNA-seq. Corresponding data for NPCs are shown in Fig S1A. **(C)** Overlap of all ISGs found at 1 h or 6 h of IFNβ treatment in ESCs and MEFs. **(D)** Uniform manifold approximation and projection for dimension reduction (UMAP) embedding of gene expression in ESCs (left) and MEFs (right) as two-dimensional representation of the transcriptome information. Each dot represents a single cell, and colors indicate IFNβ treatment duration. **(E)** Violin plots of scRNA-seq expression levels of the ISGs Ifit1 and Isg15 in single ESCs (top) and MEFs (bottom). For both genes, the number of transcripts detected largely increased in ESCs and MEFs from 1–6 h. **(F)** Expression levels of selected ISGs identified by bulk RNA-seq analysis from aggregated scRNA-seq in MEF clusters 0, 1, 2, and 3.
Source data are available for this figure.

found in NPCs mainly represented a subset of MEF ISGs (227 of 244) pointing to a high similarity of the IFNβ response in NPCs and MEFs (Fig S1B). A differential gene expression analysis of only intronic reads to assess nascent RNA levels gave very similar results with a somewhat lower number of ISGs detected in ESCs (Fig S1D and E and Table S2). We conclude that changes induced by IFNβ occurred predominantly at the gene expression level with only minor differences in RNA stability. The differences in gene induction by IFNβ were most pronounced between ESCs and MEFs, whereas NPCs showed a pattern very similar to MEFs. Accordingly, we subsequently focused on the ESC–MEF comparison to further elucidate the underlying differences.

**IFNβ response is mostly homogenous at the single-cell level**

We assessed by single-cell RNA sequencing (scRNA-seq) if the transcriptional response in ESCs and MEFs was homogeneous or if the observed up-regulation of ISGs arises from a subset of more strongly responding cells (Figs 1D and S1F). Before induction, the ESC population displayed homogeneous transcription profiles, whereas MEFs consistently formed two distinct clusters (clusters 0 and 1 and clusters 2 and 3, respectively). The separation of MEFs into two clusters arose from up-regulated genes associated with KEGG pathway "extra cellular matrix receptor interaction" in clusters 0 and 2 as opposed to the "focal adhesion" KEGG pathway in clusters 1 and 3 (Fig S1G). Based on these expression profiles we annotated clusters 0 and 2 as "mesenchymal-like" and clusters 1 and 3 as "epithelial-like."

Inspection of the UMAP plots showed no separate clustering of untreated (0 h) and 1 h IFNβ-treated ESCs, whereas they separated within the same clusters in MEFs. The differences are in line with the lower number of 57 and 115 ISGs detected by bulk RNA-seq after 1 h as compared with 188 and 452 genes after 6 h for ESCs and MEFs, respectively (Fig 1B). After 6 h stimulation, distinct clusters were present for both ESCs and MEFs. The response increase from 1–6 h is illustrated for two ISGs, Ifit1 and Isg15 in Fig 1E. We conclude that the apparent heterogeneity after 1 h of IFNβ treatment arises to a significant extend from the reduced detection sensitivity of scRNA-seq for lowly expressed genes that show an increased drop-out frequency (Yamawaki et al, 2021). The ISG expression patterns and IFNβ response dynamics of the two MEF clusters (cluster 0 versus 1 and cluster 2 versus 3) were highly similar (Fig 1F). Thus, the IFNβ response was rather homogeneous after 6 h of IFNβ treatment in the two different cell types at the single-cell level, and the ISG definition from the bulk RNA-seq analysis was used for further analysis.

**ISG expression varies between cell types in response strength and specificity**

Next, we compared the transcriptional response with IFNβ in the three cell types in further detail.

The distribution of gene expression levels in non-stimulated cells was fitted with distributions for active and repressed genes to define a background threshold for evaluation of differences in the

IFNβ response (Fig S2A). In ESCs and MEFs, some genes like Irf9, Stat1, and Stat2 were already lowly expressed in unstimulated cells and showed a significant increase in expression after IFNβ treatment (Fig 2A). Other ISGs like Irf7, Rtp4, and Usp18 changed from repressed to active after IFNβ stimulation. Compared with ESCs, MEFs displayed a 10–100-fold stronger induction of these common ISGs, which is in line with previous findings (Wang et al, 2014). To further dissect the overall stronger response in MEFs, we compared the expression levels of factors of the IFN signaling pathway. The Ifnar1 and Ifnar2 receptors and Jak1 kinase were higher expressed in MEFs than in ESCs, whereas for key transcription factors Stat1, Stat2, and Irf9, no differences were identified (Fig S2B). A Western blot with STAT1 and STAT2 antibodies showed that STAT1 and STAT2 proteins were present at lower levels in ESCs before and after IFNβ induction than MEFs (Fig 2B).

The amount of STAT1 phosphorylated at residue 701 ($STAT1_{p701}$) or 727 ($STAT1_{p721}$) was clearly increased after 1 h in MEFs as compared with ESCs and decayed to low levels at the 6 h time point. Furthermore, lower levels of active STAT1/2 protein complexes upon IFNβ induction are apparent from comparing the amounts of $STAT1_{p701}$ and $STAT1_{p727}$ between ESCs and MEFs. These differences in STAT1/2 protein levels under stimulated and unstimulated conditions were not reflected in the RNA levels and suggest a reduced protein translation/degradation ratio of STAT1 and STAT2 in ESCs. We conclude that the globally attenuated response to IFNβ in ESCs involved epigenetic networks that lead to a reduced activity of key components of the JAK/STAT signaling pathway both on the RNA and protein level as compared with differentiated cells.

Cell type–specific differences were apparent as illustrated for selected genes in Fig 2C. After 6 h of stimulation, Ccnd2, Ifi27, and Nsg2 were induced in ESCs. In MEFs, expression of all three genes was not up-regulated. In contrast, Ccl2, Gbp6, and Ifit1bl1 were specifically up-regulated in MEFs upon IFNβ stimulation. Gbp6 was lowly induced in ESCs but only after 6 h. In summary, large cell type–specific differences in gene expression levels were observed upon IFNβ stimulation between the three cell types that involved the expression of distinct sets of ISGs.

**STAT1/2 binding is cell type specific and correlates with ISG activation**

The differences in IFNβ response raise the question why certain ISGs were preferably expressed in one cell type and not in the other. To reveal molecular details of gene expression regulation, we mapped STAT1p701 and STAT2 binding by chromatin immunoprecipitation after sequencing (ChIP-seq). Antibodies against $STAT1_{p701}$ and STAT2 in ESCs and MEFs were used, and exemplary regions enriched for both transcription factors are shown in Fig 3A. Peaks detected in ESCs and MEFs after 1 and 6 h of IFNβ treatment (Tables S2 and S4) were combined to create one common list for all downstream analysis. A total of 208 peaks in ESCs and 276 peaks in MEFs were bound simultaneously by both transcription factors (Fig 3B and Tables S2 and S4). These loci were annotated as "STAT1/2" binding sites in our analysis. They are likely to represent the ISGF3 complex as it has been shown previously that STAT1 and STAT2 assemble with IRF9 to form the ISGF3 complex upon IFN stimulation (Platanitis et al, 2019). A total of 392 STAT1/2 binding

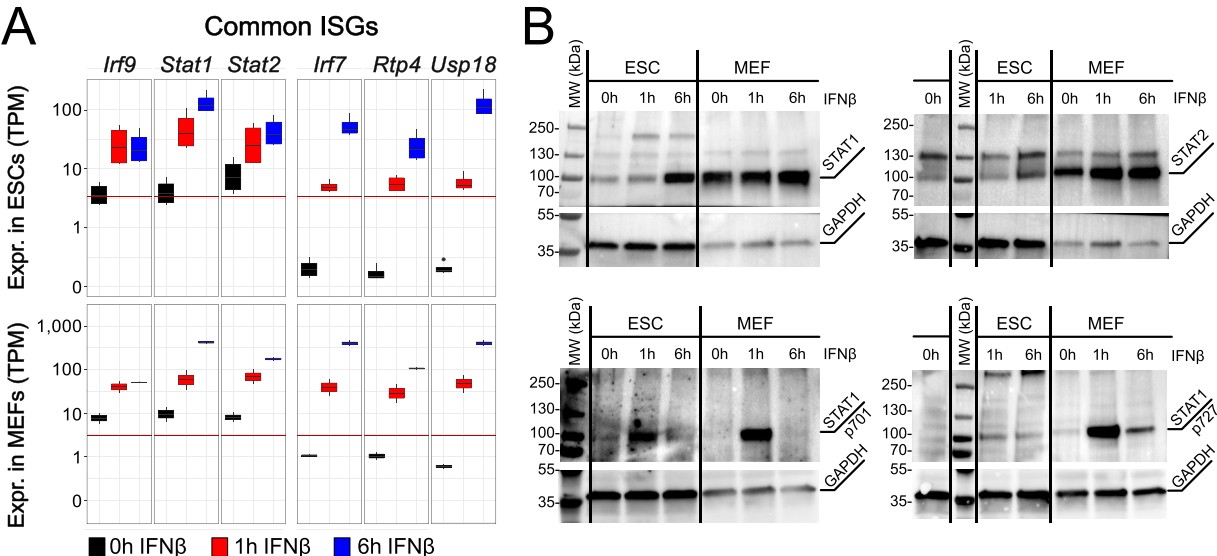

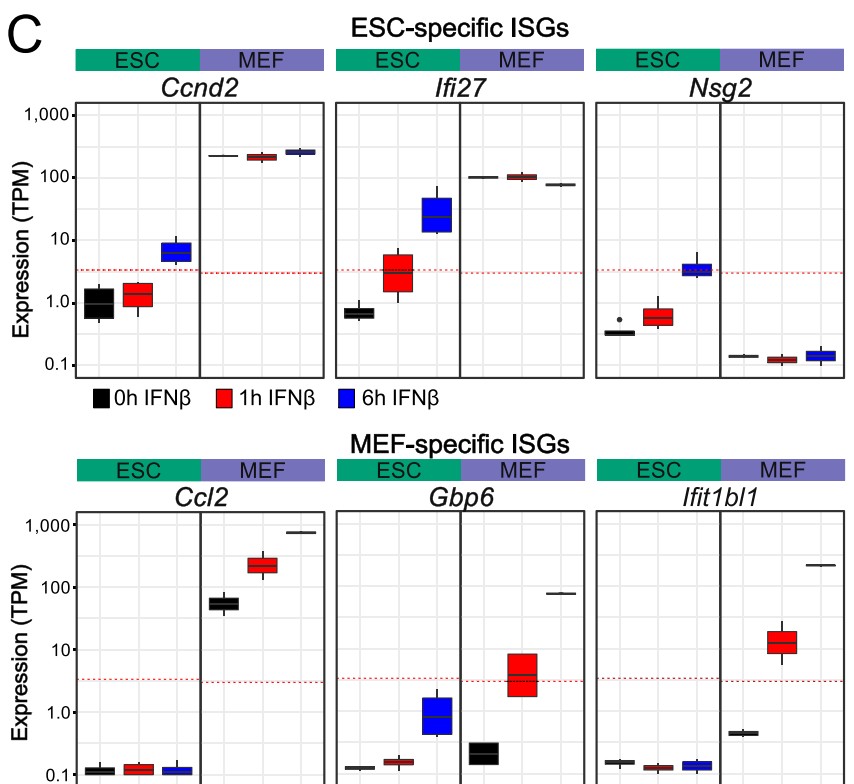

**Figure 2. Cell type–specific ISG induction and protein expression.**
**(A)** Normalized gene expression levels of selected ISGs from bulk RNA-seq in ESCs (top, n = 4) and MEFs (bottom, n = 2). Gene expression is given as transcripts per kilobase million (TPM). **(B)** Western blots of IFNβ-stimulated ESCs and MEFs at 0, 1, and 6 h time points. The top row shows total levels of STAT1 (left) and STAT2 (right). The lower row shows phosphorylation of STAT1 at position 701 (left) and 727 (right). GAPDH was used as a housekeeping gene control. **(C)** Normalized gene expression levels from bulk RNA-seq of selected cell type–specific ISGs in ESCs (n = 4) and MEFs (n = 2). The red line represents a cell type–specific threshold to distinguish active and repressed genes. Top: expression of ISGs Ccnd2, Ifi27, and Nsg2 was only induced in ESCs. Bottom: expression of ISG *Ccl2*, *Gbp6*, and *Ifit1bl1* was induced in MEFs. Source data are available for this figure.

sites were determined from the combined data set of ESCs and MEFs after 1 and 6 h of IFNβ stimulation. The remaining peaks that only had STAT1p701 or STAT2 bound were classified as "STAT1" and "STAT2" binding sites, respectively. The overlap of peaks between cell types was moderate (Fig 3E). Only 38 sites were found to be bound by STAT1 in both cell types, whereas most STAT2 peaks were

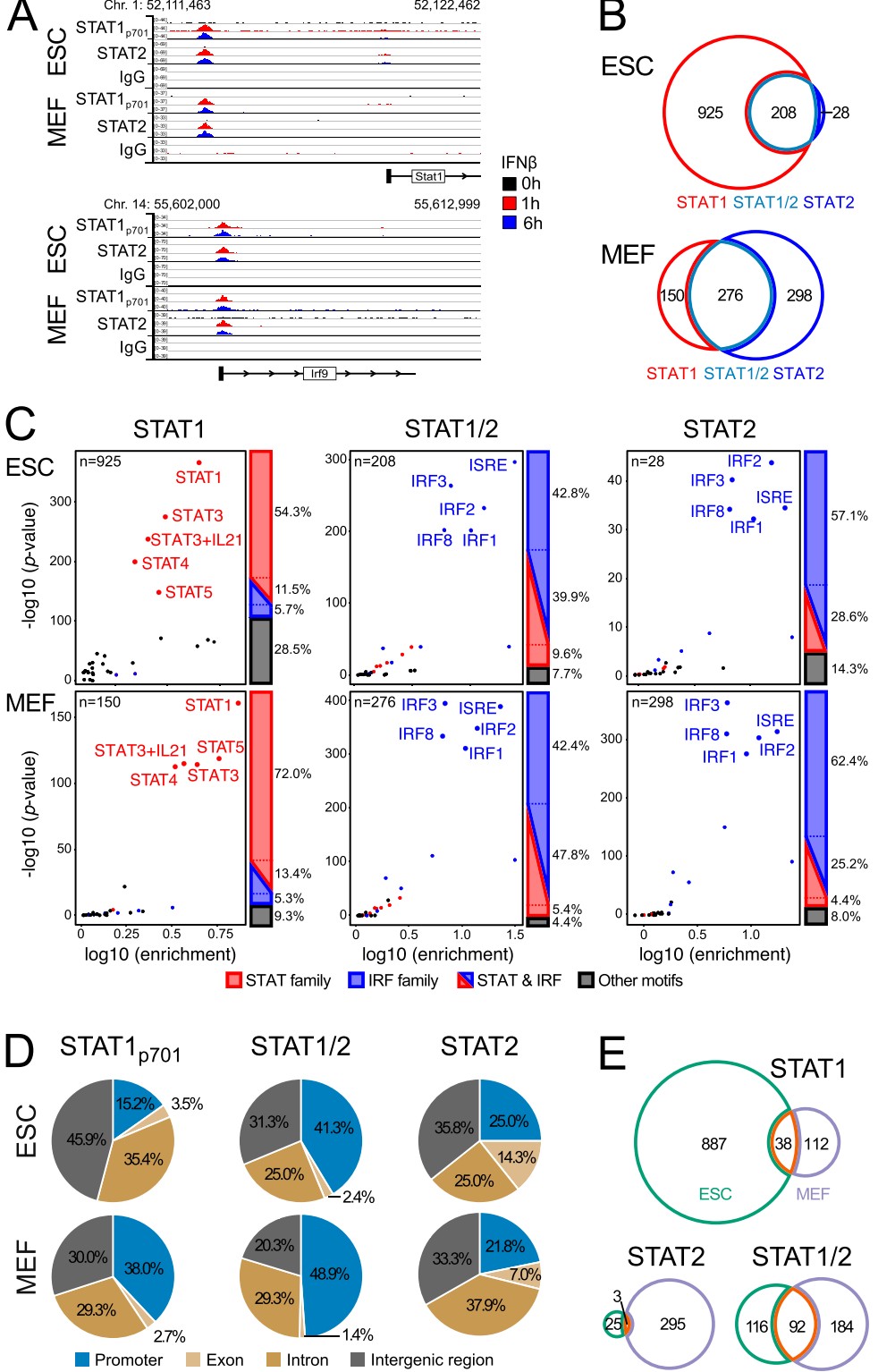

**Figure 3. Binding of STAT1 and STAT2 in ESCs and MEFs.**
**(A)** ChIP-seq of STAT1$_{p701}$ and STAT2 upstream of *Stat1* (top) and at the *Irf9* promoter (bottom). Tracks show one replicate for each condition. **(B)** STAT1$_{p701}$ and STAT2 peaks in ESCs and MEFs. The STAT1/2-binding sites were defined by the overlap of STAT1$_{p701}$ and STAT2 peaks from the combined list of peaks detected at 1 and 6 h of IFNβ treatment. Sample numbers are given in Table S1. **(C)** Enrichment of transcription factor binding motifs in STAT1p701, STAT1/2, and STAT2 peak sets identified in ESCs and MEFs. Motif color scheme: STAT-family (STAT1, STAT3, STAT3+IL21, STAT4, and STAT5), red; IRF-family (IRF1, IRF2, IRF3, IRF8, and ISRE [IRF9]), blue; other, black. Four biological replicates for ESCs and two for MEFs were analyzed. **(D)** Distribution of STAT1p701, STAT1/2, and STAT2 peaks at all annotated promoters, exons, introns, and intergenic regions annotated from the Ensembl database. **(E)** Overlap of STAT-binding sites between ESCs and MEFs for STAT1p701, STAT1/2, and STAT2.
Source data are available for this figure.

cell type specific. STAT1/2-binding sites common to both cell types comprised 44% (ESC) and 33% (MEFs) of the peaks. To validate the peak specificity, we determined enriched known motifs in STAT-binding sites. In both ESCs and MEFs, the STAT-family motifs

(STAT1, STAT3, STAT3 + IL21, STAT4, and STAT5) were enriched at STAT1 peaks, whereas IRF-family motifs (IRF1, IRF2, IRF3, IRF8, and ISRE) were the most enriched motifs in the STAT1/2 and STAT2 peaks (Fig 3C). Within each family, the motifs were highly similar

(Fig S3A). At least one of these family motifs was found in 66% (STAT1), 83% (STAT1/2), and 86% (STAT2) of the ESC peaks and 85% (STAT1), 90% (STAT1/2), and 88% (STAT2) of the MEF peaks. Thus, the same motifs were recognized independent of cell type and in line with the classification into STAT1-, STAT2-, and STAT1/2-binding sites. This conclusion was corroborated by a de novo motif analysis (Fig S3B and C). The top de novo motif was in all groups in one of the STAT or IRF families with a similarity score of ~0.9. It is noted that the total number of the 1,885 STAT peaks detected by ChIP-seq represents only a minor fraction of the ~2.5 million STAT- or IRF-family sequence motifs in the mouse genome (~0.8 million IRF motifs and 1.7 million STAT motifs were extracted from the HOMER database) (http://homer.ucsd.edu/homer/index.html) (Heinz et al, 2010). Based on these findings, we conclude that the DNA sequence is neither sufficient to predict the experimentally observed STAT-binding sites nor can it rationalize the differences in STAT-binding sites detected between cell types.

**ISG activation can be partly assigned to STAT promoter binding**

To further dissect the activation mechanism, we analyzed the spatial relation between STAT-binding sites and ISGs. Almost half of the STAT1/2 peaks in ESCs and MEFs were located at promoters (defined as a window of ±1 kb around the transcription start site) with around 3/4 of them at the ISGs identified from the bulk RNA-seq analysis (Figs 3D and S3D and Table S5). In contrast, a smaller fraction of 15–38% of the STAT1 or STAT2 only peaks was at promoters. In addition, the promoters that displayed STAT1 binding but lacked STAT2 were mostly highly expressed genes. Only a minor fraction of 6% in ESCs and 16% in MEFs was at ISG promoters, although this fraction was around 50% for the STAT2 only peaks. Based on this analysis we conclude that STAT1/2 binding (representing bona fide ISGF3 complexes together with IRF9) at promoters was the main driver of ISG activation in our system ($n$ = 71 in ESCs; $n$ = 112 in MEFs). In addition, ISG activation was provided for a smaller fraction of promoters by STAT2 in the absence of STAT1 ($n$ = 5 in ESCs; $n$ = 34 in MEFs). The latter finding is in line with the conclusion that the STAT2–IRF9 complex alone could provide some activation (Platanitis et al, 2019). STAT1 without STAT2 appeared to lack significant activation capacity in our system but rather displayed some propensity to bind to already active promoters. Nevertheless, it could potentially be involved in promoting transcription of some ISGs where it was found at the promoter ($n$ = 10 in ESCs; $n$ = 11 in MEFs). For a remaining fraction of 105 (ESCs) and 306 (MEFs) ISGs, no STAT binding at the promoter was detected. Accordingly, these ISGs were either secondary target genes or become activated from non-promoter STAT-binding sites. Based on these findings, we focused on STAT1/2-binding sites as a proxy for the ISGF3 complex to further characterize the relation between non-promoter STAT1/2 binding and ISGs.

**STAT1/2-driven enhancers are predicted from co-accessibility analysis**

The non-promoter STAT1/2 peaks could represent enhancer elements that regulate ISGs from a distance. A simple assignment of these potential enhancer sites to the nearest gene linked them to only a small fraction ISGs that lacked a promoter bound STAT1/2 complex ($n$ = 13 in ESCs; $n$ = 41 in MEFs) (Fig S4A and C). Thus, the assumption that most enhancer targets can be predicted by selecting the closest gene is not justified in our system. To further characterize potential targets of STAT1/2 binding at putative enhancers, we inferred links to ISG promoters from a co-accessibility analysis of the single-cell ATAC sequencing (scATAC-seq) data (Figs 4A–C and S4B and Table S6). The corresponding UMAP plot showed no clear separation of ESCs and MEFs before and after IFNβ treatment (Fig 4A), indicating that the gain in chromatin accessibility at STAT1/2 sites was not accompanied by a global alteration of the chromatin landscape (Fig S4B). These observations agree with the number of ISGs identified by RNA-seq that represent only a small fraction of the total transcriptome. For MEFs, two separate cell clusters were also present in the scATAC-seq data (Fig 4B) and assigned to epithelial- and mesenchymal-like MEF subtypes by integration with the scRNA-seq data (Figs 1D and 4C). Next, we computed correlations between pairs of genomic loci that were simultaneously accessible in the same cell based on previously described approaches (Mallm et al, 2019; Granja et al, 2021) to reveal links between enhancers with STAT1/2 binding and ISGs (Table S5). This analysis was conducted for all 392 STAT1/2-binding sites in ESCs and MEFs in a 1 Mb window.

As an exemplary result, STAT1/2 binding to a putative distal enhancer in ESCs is depicted in Fig 4D for the *Uba7* ISG. An IFNβ-induced co-accessible link between the STAT1/2 bound enhancer candidate and the promoter of the ISG *Uba7* was detected. Another example of ISG regulation by STAT1/2 binding to distal putative enhancers is shown for the Ly6 gene cluster in MEFs (Fig 4E). Expression of ISGs *Ly6e*, *Ly6a*, and *Ly6c1* increased with IFNβ treatment in mesenchymal- and epithelial-like MEFs. In the pseudo-bulk ATAC-seq data the promoter of ISG *Ly6e* was highly accessible at different time points, whereas *Ly6a* and *Ly6c1* promoters remained in lower accessible states. Upon IFNβ treatment, multiple co-accessible links between three intergenic STAT1/2 sites and ISGs were detected, either directly to the *Ly6* promoters or indirectly to their gene bodies or proximal regions. The changes involved the formation of new links between the potential enhancer cluster and the *Ly6a* and *Ly6c1* promoters and the loss of links present at the 0 h time point. In line with the different pseudo-bulk accessibility profiles, the observed combinatorial co-accessible links between STAT1/2 sites and ISGs varied between the two MEF subtypes. By applying this analysis to all STAT1/2-binding sites identified, we were able to link ~25% of ISGs without the STAT1/2 promoter binding to a distal STAT1/2 binding event after IFNβ induction (Fig 4F) (ESCs, 27 ISGs; epithelial-like MEFs, 84 ISGs; and mesenchymal-like MEFs, 85 ISGs) (Table S5). Interestingly, we also observed a loss of existing co-accessible links between ISGs and distal STAT1/2 sites at several loci (ESCs, 10 ISGs; epithelial-like MEFs, 14 ISGs; and mesenchymal-like MEFs, 16 ISGs), which points to larger changes of the 3D chromatin organization during activation that could involve the resolution of inhibitory interactions. These regulatory mechanisms of ISG induction by STAT1/2 binding are not mutually exclusive. For ~20% of ISGs, we observed more than one mechanism (Fig S4C) (ESCs, 46 ISGs; epithelial-like MEFs, 75 ISGs; and mesenchymal-like MEFs, 97 ISGs). Moreover, we were able to differentiate

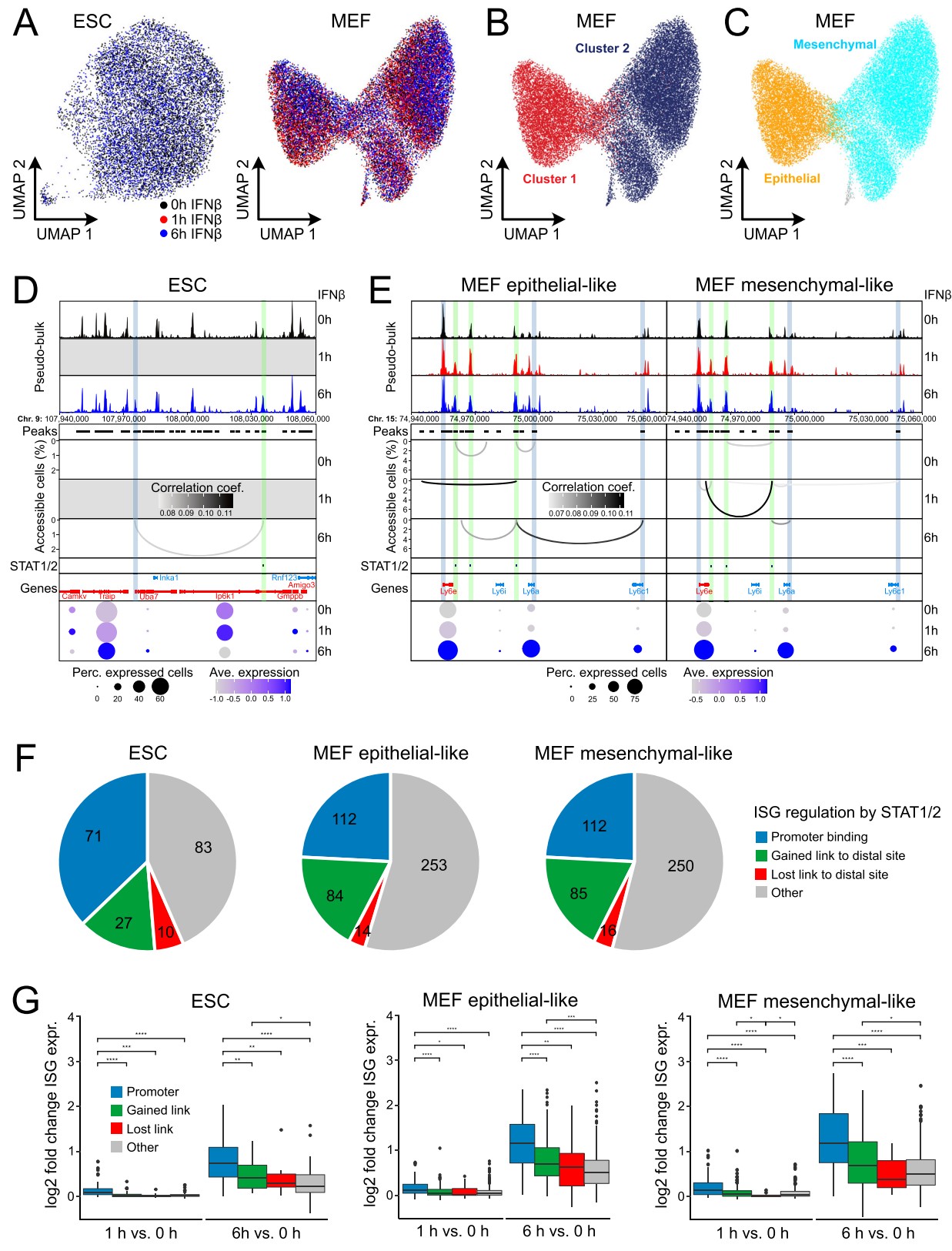

**Figure 4. Regulation of ISG expression by distal STAT1/2 binding.**
**(A)** UMAP embedding of chromatin accessibility in ESCs (left) and MEFs (right). Each dot represents one cell and is colored according to treatment. **(B)** Same as panel (A) for MEFs with single cell coloring according to k-nearest neighbor clusters. **(C)** Same as panel (B) with single cell coloring according to MEF subtypes derived from scRNA-seq data integration by gene activities. **(D)** Co-accessibility before and after 6 h of IFNβ induction of ESCs in a region around the Uba7 ISG. Top: browser tracks of aggregated

co-accessible links from ISG promoters to distal STAT1/2-binding events either in other ISG promoters (75%) or in exonic, intronic, or intergenic enhancers (25%) (Fig S4D).

## Binding of STAT1/2 to distal sites correlates with efficient target ISG induction

Next, we investigated the expression of differently regulated ISG categories after 1 and 6 h of IFNβ treatment over unstimulated control cells and found similar patterns for ESCs and both MEF subtypes (Fig 4G). After 1 h of IFNβ treatment, induction was relatively low, but after 6 h a clear expression increase was observed for all cell types and ISG categories. ISGs with STAT1/2 bound at their promoter had the strongest expression up-regulation. Nevertheless, ISGs that gained a co-accessible link to a distal STAT1/2 bound site showed a significantly stronger expression induction after 6 h of IFNβ treatment compared with ISGs without any link to STAT1/2. Interestingly, ISGs that lost a preexisting link to a distal STAT1/2 bound site upon IFNβ treatment showed a significantly lower gene expression level before IFNβ treatment (0 h) in ESCs and mesenchymal-like MEFs in support of inhibitory interactions before induction (Fig S4E). In summary, the scATAC-seq data allowed us to distinguish different mechanisms by which STAT1/2 binding regulates ISG expression. It identifies a significant number of ISGs that appear to be regulated by STAT1/2 binding to distal enhancers in addition to those with direct binding of the activator to the promoter. Moreover, our analysis suggests that the loss of preexisting long-range interactions during STAT1/2 binding could be associated with the removal of inhibitory interactions.

## Five different chromatin states of STAT1/2-binding sites can be distinguished

The overlap of STAT1/2 peaks from ESCs and MEFs revealed 92 shared binding sites mostly at promoters (70/92). The 116 ESC-specific and 184 MEF-specific sites were predominantly at non-promoter loci (100/116 and 118/184) (Fig S5A). We reasoned that the cell type–specific STAT1/2 binding was dependent on the chromatin context. Accordingly, we mapped six histone modifications (H3K4me1, H3K4me3, H3K9ac, H3K27ac, H3K9me3, and H3K27me3) by ChIP-seq and chromatin accessibility by ATAC-seq. Exemplary regions for ESCs and MEFs were shown (Fig 5A). STAT1/2 binding at the *Usp18* promoter induced the gene in both cell types from a transcriptionally repressed to an active state. In contrast, *Ifi27* was induced in ESCs as compared with a constitutively active state in MEFs, whereas *Gbp6* became active in MEFs and remained silent in ESCs. Of note, several additional ISRE motifs did not display STAT1/2

binding, which illustrates the requirement for a permissive chromatin state. To reveal chromatin features that are linked to STAT1/2 binding, normalized read counts in a window of ±1 kb around the peak center were computed for the different readouts (Fig S5A). These data were then subjected to unsupervised *k*-means clustering (Figs 5B and S5B and C). Five main clusters emerged that were annotated based on the combination of enriched chromatin features (Fig 5B): (i) "Active Promoter" was enriched for H3K4me3, H3K9ac, and H3K27ac (Ernst et al, 2011). (ii) "Active Enhancer" was marked by H3K4me1 and H3K27ac (Creyghton et al, 2010). (iii) The "Bivalent" state carried active marks like H3K4me3 and repressive marks like H3K27me3 at the same time (Bernstein et al, 2006). (iv) The "Poised" state showed only H3K4me1 (Creyghton et al, 2010). (v) "Repressed" was marked by enrichment of H3K9me3 or H3K27me3 (Lehnertz et al, 2003; Morey & Helin, 2010).

## STAT1/2 binding is directed by chromatin accessibility and specific histone marks

Next, chromatin states at STAT1/2 sites in ESCs and MEFs and their changes were analyzed (Fig 5C–E). The most pronounced chromatin state differences between cell types were between the "Poised" and "Repressed" states in ESCs and the "Active Enhancer," "Bivalent," and "Poised" states in MEFs (Fig 5C). The 116 ESC-specific sites displayed a three to fourfold loss of the "Active Promoter" and "Active Enhancer" states and an ~fivefold increase of the "Repressed" state as compared with the chromatin state of these sites in MEFs (Fig 5E). Corresponding changes of the "Active Enhancer" and "Repressed" states were also found for MEF-specific sites in ESCs and MEFs. The fraction of MEF-specific STAT1/2 sites in the "Active Promoter" state remained mostly unchanged between cell types, whereas the number of sites in the "Bivalent" state strongly increased from 3 to 56 sites (Fig 5E). We conclude that the main differences that determine the cell type–specific binding of STAT1/2 occurred between the "Repressed" state (H3K9me3 and H3K27me3) and "Active Enhancer" and "Bivalent" states that both are enriched in the H3K4me1 and H3K27ac modifications. Accordingly, the increased number of ISGs detected in MEFs appears to be related to the more frequent activation of ISRE-containing enhancer elements.

To further dissect the relation between chromatin signals in the uninduced state and STAT1/2 binding upon induction, we computed their correlations. Normalized read counts of a given chromatin feature before induction were plotted against STAT1/2 binding as represented by the average signal of STAT1 and STAT2 at 1 h of IFNβ treatment at the same locus (Fig 6A). These plots visualized the differences between ESC-specific (black) and MEF-specific (red) binding sites for the indicated chromatin features. The *P*-value and

---

pseudo-bulk chromatin accessibility from single cells. Middle: co-accessible links between the indicated intronic STAT1/2 bound site 371 (differential STAT1/2 peak after 1 h of IFNβ treatment in ESCs) and other genomic loci. Experimentally identified ISG promoters (blue) and sites with bound STAT1/2 after 1 and/or 6 h (green) are marked. Bottom: gene expression levels from scRNA-seq. Transcription from Inka1 and Rnf123 was not detected. **(E)** Same as panel (D) but for three intergenic STAT1/2 bound sites 125, 126, and 127 (differential STAT1/2 peaks after 1 and 6 h of IFNβ treatment in MEFs) in the Ly6 ISG cluster in MEFs. **(F)** ISG regulation mechanisms according to STAT1/2 binding after IFNβ treatment. Promoter bound STAT1/2 (independent of the presence of additional links to distal sites), blue; gained co-accessible link to a distal STAT1/2 peak, green; ISGs that lost a co-accessible link to a distal STAT1/2 peak after IFNβ treatment, red; other ISGs, grey. **(G)** Expression changes of ISGs for the different STAT1/2-dependent regulation types shown in panel (F) from bulk RNA-seq data. *P*-values from a Wilcoxon rank-sum test are indicated as *, $P < 0.05$; **, $P < 0.01$; ***, $P < 0.001$; ****, $P < 0.0001$.
Source data are available for this figure.

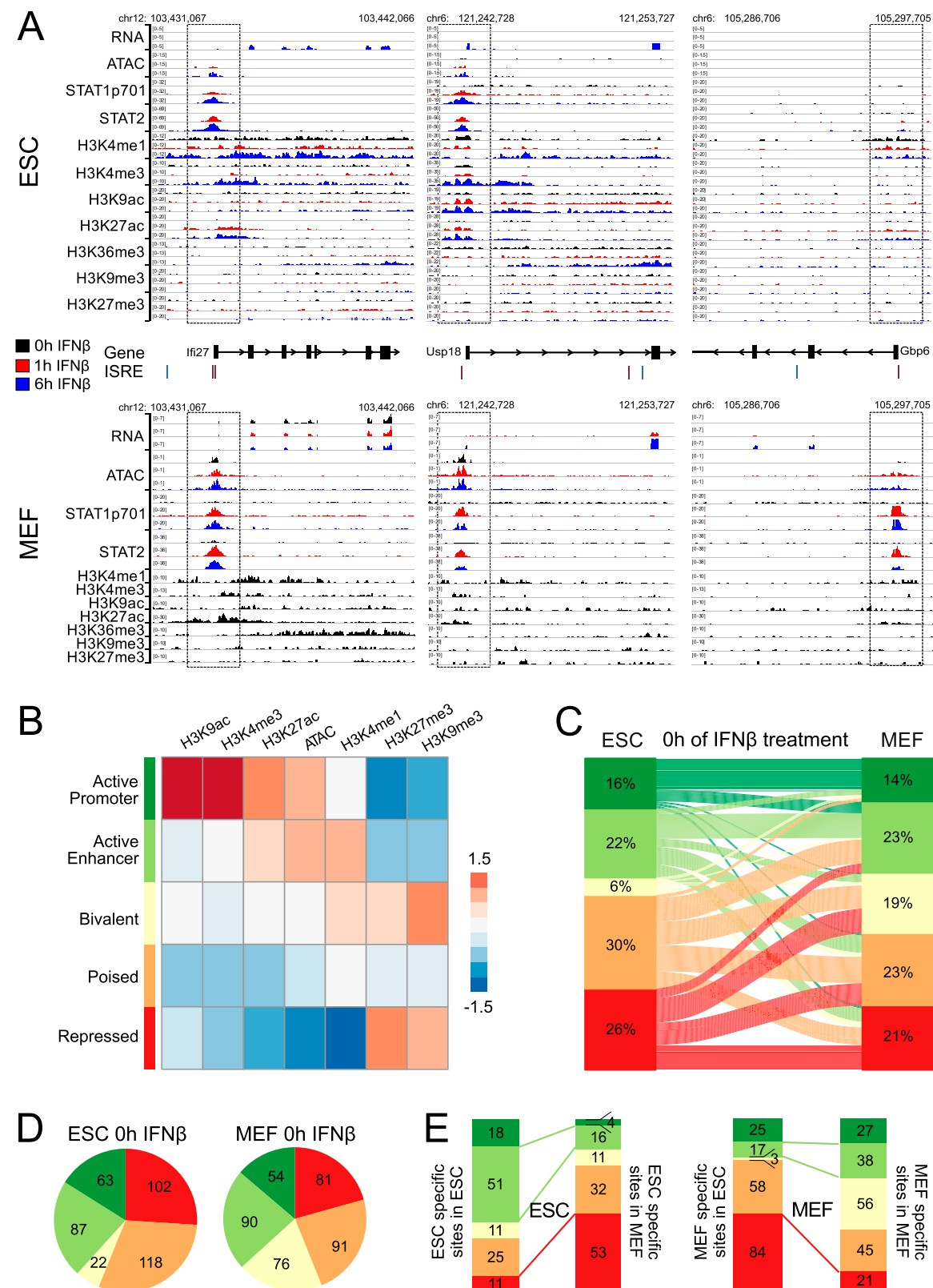

**Figure 5. Contribution of chromatin features to STAT1/2 binding.**
**(A)** Genomic regions around the ISGs *Ifi27*, *Usp18*, and *Gbp6* in ESCs (top) and MEFs (bottom) with the different sequencing readouts and the promoter regions marked by boxes. Gene annotation was based on Ensembl, and the positions of the DNA binding motif IRSE were extracted from the HOMER database. Each browser track shows one representative biological replicate. **(B)** Heatmap of unsupervised k-means clustering of histone modifications and ATAC data at 392 STAT1/2-binding sites. The indicated

correlation coefficient $R$ of a given mark with STAT1/2 binding are plotted in Fig 6B. ATAC (ESC, $R$ = 0.42; MEF, $R$ = 0.53), H3K4me1 (ESC, $R$ = 0.45; MEF, $R$ = 0.43), and H3K27ac (ESC, $R$ = 0.41; MEF, $R$ = 0.63) were the most strongly positively correlated marks, whereas H3K27me3 (ESC, $R$ = −0.23; MEF, $R$ = −0.39) was anticorrelated with STAT1/2 binding. For the repressive H3K9me3 mark, the correlation was negative for ESCs ($R$ = −0.26) and slightly positive for MEFs ($R$ = 0.08) pointing to a more complex relation. We concluded that a preexisting active chromatin state (open chromatin, H3K4me1, and H3K27ac) promoted STAT1/2 binding, whereas chromatin loci marked by the H3K27me3 impeded binding (Fig 6C).

## Discussion

Our genome-wide multi-omics comparison of ESCs and MEFs reveals mechanisms that govern the cell type–specific response to IFN$\beta$. In total, 513 ISGs were identified in line with previous studies that reported between 200 and 1,000 up-regulated genes in different cellular systems (Der et al, 1998; de Veer et al, 2001; Mostafavi et al, 2016). Our results corroborate the finding that ESCs show an attenuated response to IFN$\beta$ (Whyatt et al, 1993; Gonzalez-Navajas et al, 2012; Wang et al, 2013; Wang et al, 2014; Guo et al, 2015; D'Angelo et al, 2016; Guo, 2017). This stem cell–specific feature appears to be compensated by a constitutive expression of some ISGs in human stem cells (Wu et al, 2018) and the presence of an antiviral RNA interference-based system in mouse ESCs (Maillard et al, 2013; Poirier et al, 2021). Furthermore, ESC chromatin displays global differences in epigenetic modification patterns and chromatin accessibility as compared with differentiated cells (Lim & Meshorer, 2021). These features are linked to their self-renewal capacity and pluripotency and an associated specific activity of several signaling pathways, which includes an attenuated IFN-I system response (Kristensen et al, 2005; Eggenberger et al, 2019; Gordeeva, 2019).

A previous RT-qPCR analysis of selected components of the IFN signaling pathway in ESCs identified a significant down-regulation of the IFN$\alpha$/$\beta$ receptor Ifnar1, whereas Stat2, Tyk2, and Irf9 were up-regulated as compared to a MEF cell line (Wang et al, 2014). Based on our differential RNA-seq maps of the unstimulated cell types, we confirm the down-regulation of Ifnar1 in ESCs while the differences for Stat2, Tyk2, and Irf9 were above the $P$ < 0.01 significance level. We additionally detected a strong down-regulation of Ifnar2, Ifngr1/2, and Jak1/2 kinases in ESCs relative to NPCs and MEFs. Thus, a globally reduced IFN$\beta$ response could be assigned to lower expression of key components of the IFN-signaling pathway in ESCs. In addition, both STAT1 and STAT2 and phosphorylated STAT1 were more abundant in MEFs than in ESCs at the protein level. These differences in STAT protein abundance were not reflected in the RNA levels. The gene-specific correlation between the RNA and protein levels can vary

three orders of magnitude but relatively large variations for the same protein between different cell types are unusual (Edfors et al, 2016). The results obtained here for STAT proteins could be related to cell type–specific differences in the regulation of STAT1 and STAT2 protein degradation that is dependent on the STAT phosphorylation state in a complex manner (Kok et al, 2020; Lee et al, 2020).

Previous studies on STAT1/2 binding reported 6,703 STAT2 peaks for IFN$\alpha$-treated B cells (Mostafavi et al, 2016), and 41,582 (IFN$\gamma$-stimulated) and 11,004 (unstimulated) STAT1-binding sites in HeLa S3 cells (Robertson et al, 2007). The specificity of STAT peak assignment in these previous studies appears to be moderate. A fraction of 46% of the STAT2 peaks displayed a >twofold increase upon IFN$\alpha$ treatment (Mostafavi et al, 2016), whereas a two to fivefold enrichment of GAS and ISRE sequences in the STAT1 peaks was present (Robertson et al, 2007). Our identification of STAT1p701- and STAT2-binding sites was more stringent and involved an at least fourfold increase upon induction with a similar number of 208 and 276 STAT1/2 peaks in ESCs and MEFs. In addition, 80–90% of the STAT-binding sites carried a STAT- or IRF-family sequence motif with more than 10-fold higher frequency than that found in the background sequences. It is further noted that we did not detect STAT2 ChIP-seq peaks before the IFN$\beta$ stimulus. Thus, an activity of unphosphorylated STAT2/IRF9 for basal gene expression of ISGs as reported in Blaszczyk et al (2015) and Platanitis et al (2019) was not apparent in the binding site maps recorded here. For example, the Ly6e promoter was previously identified as a STAT2/IRF9 target that assembled into an ISRF complex (Platanitis et al, 2019), whereas our data suggest that activation in MEFs occurs via a downstream STAT1/2 enhancer (Fig 4E). In general, the differences between studies with respect to STAT-binding sites and ISGs are likely to reflect on the one hand cell type–specific features. On the other hand, also the IFN$\beta$ concentration and treatment duration and the experimental and data analysis methods to call ISGs and STAT peaks will affect the results. For example, differences in the ChIP-seq protocols with respect to the antibodies used or the method of chromatin fragmentation and the peak calling method are known to change the binding site maps (Kidder et al, 2011).

The main ISG activation sites in our system had STAT1 and STAT2 bound simultaneously, most likely within the ISGF3 complex that additionally involves IRF9 and in line with previous findings (Stark & Darnell, 2012; Ivashkiv & Donlin, 2014; Chen et al, 2017; Villarino et al, 2017; Au-Yeung & Horvath, 2018; Hu et al, 2021). This assignment was confirmed by the binding motif analysis that yielded an enrichment of IRF motifs in 80–90% of the 392 STAT1/2 peaks. The number of sites that had only STAT1$_{p701}$ or STAT2 bound was 1,037 and 323, respectively (Table S2). STAT1 homodimers can also act as activators of type I IFN response (Stark & Darnell, 2012; Stanifer et al, 2019). However, the promoters that only had STAT1$_{p701}$ bound showed no enrichment for ISGs in our data set. An additional minor contribution to ISG activation arose from binding of STAT2 in the absence of STAT1 at ISG promoters,

five main chromatin states were identified. Data from unstimulated ESCs and MEFs and ESCs, at 1 and 6 h IFN$\beta$ treatment were used. Sample numbers are given in Table S1. **(C)** Chromatin state comparison between untreated ESCs and MEFs at STAT1/2-binding sites based on the data in panel (B) and corresponding coloring of the five different chromatin states. The lines link the same binding sites between conditions and do not represent a differentiation path between ESCs and MEFs. **(D)** Absolute numbers of STAT1/2-binding sites according to chromatin states in unstimulated ESCs and MEFs. **(E)** Distribution of 116 ESC-specific and 184 MEF-specific STAT1/2-binding sites according to the chromatin state.
Source data are available for this figure.

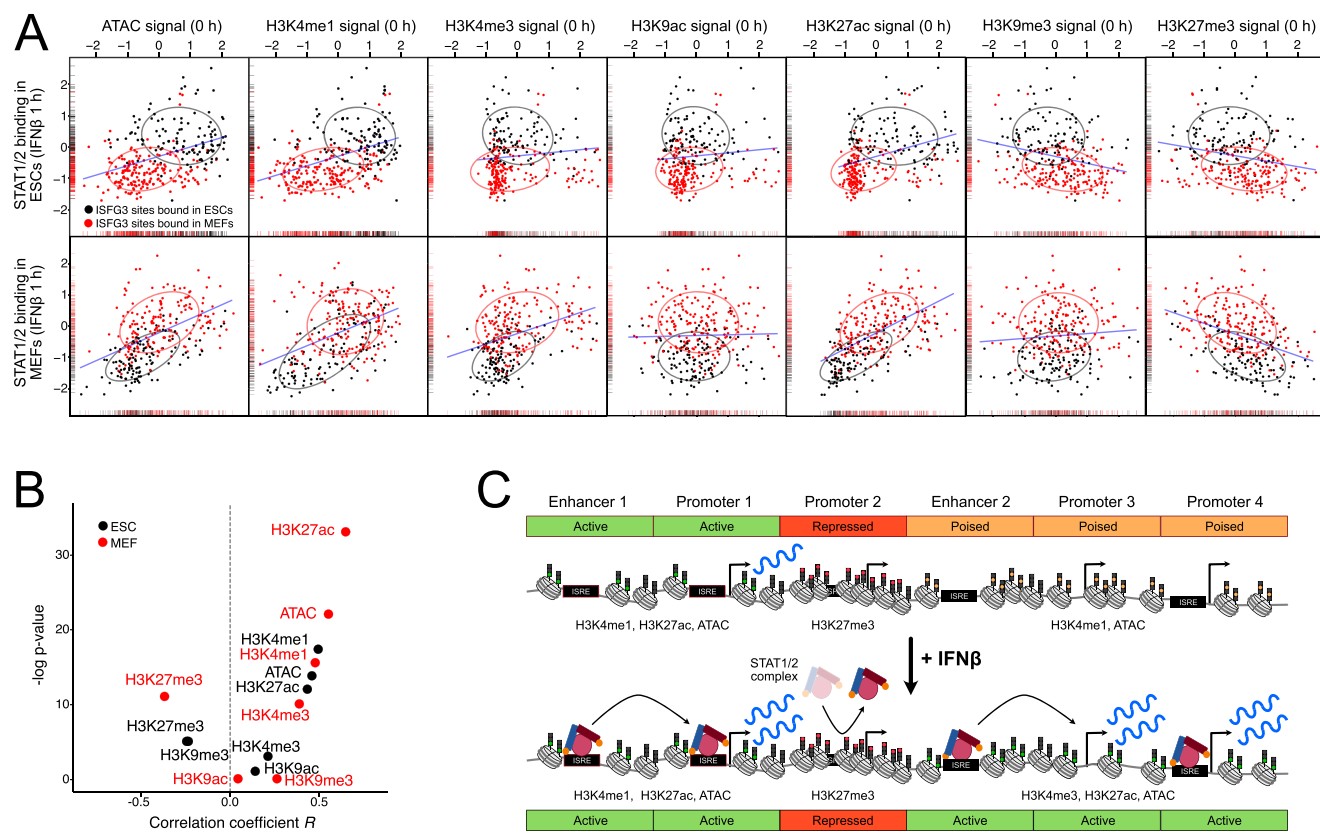

**Figure 6. Correlation of STAT1/2 binding with preexisting chromatin features.**
**(A)** Correlation between STAT1/2 binding after 1 h of IFNβ treatment and preexisting chromatin features before IFNβ treatment. The STAT1/2-binding signal was computed as the average signal of STAT1 and STAT2 after 1 h IFNβ treatment in ESCs (top) and MEFs (bottom). The chromatin features were quantified by counting the normalized read counts at the STAT1/2-binding sites before induction. ESC-specific STAT1/2-binding sites are shown in black and MEF-specific ones in red. Ellipses indicate the area, in which 75% of all data points are located. Density distributions are shown along the x- and y-axis. The blue line shows the linear regression of the combined set of ESC- and MEF-specific STAT1/2-binding sites. Sample numbers are given in Table S1. **(B)** Correlation between STAT1/2 binding and chromatin features determined for the data in panel (A) in ESCs (black) and MEFs (red). **(C)** Scheme of ISG induction via chromatin context-dependent STAT1/2 binding. Binding to ISREs at promoters or enhancers can be facilitated or repressed via preexisting chromatin states marked by the indicated chromatin features that can differ between cell types. ISGs that lack an ISRE and STAT1/2 binding at the promoter can also be activated by STAT1/2 binding to distal enhancers.
Source data are available for this figure.

which is in line with the observation that the STAT2-IRF9 complex has some activation capacity without STAT1 (Platanitis et al, 2019).

More than 2/3 of the STAT1/2 peaks were located at the intergenic or intronic regions, and thus represent potential enhancer elements that could drive ISG activation. Many of these STAT1/2 sites showed different chromatin states between ESC and MEFs, which points to their role for a cell type–specific IFNβ response (Fig 5B–E). The target genes of these putative enhancers were not necessarily those at closest genomic distance (Fig S4A). By conducting a co-accessibility analysis of the scATAC-seq data, we were able to link 25% of ISGs lacking STAT1/2 at the promoter to a distal STAT1/2 bound site. These correlated accessibilities could originate from direct spatial contacts or other mechanisms of enhancer–promoter communication (Karr et al, 2022). They are in line with a recent study that describes the reorganization of the 3D genome around ISG loci upon both IFNβ and IFNγ treatments, which involves loop formation, nucleosome remodeling, and an increase of DNA accessibility (Platanitis et al, 2022). For 75% of the ISG promoters with a co-accessibly link, the distal STAT1/2 bound site was at another ISG promoter, whereas the remaining 25% were at a bona fide enhancer. This points to a co-regulation between ISG promoter elements that is supported by a recent report showing that also promoters can act as enhancers to drive expression of ISG clusters (Santiago-Algarra et al, 2021). In addition, our data suggest that IFNβ induction and STAT1/2 binding could also involve the removal of preexisting inhibitory links between ISGs and distal regulatory regions. The latter process might be related to the loss of long-range interactions observed during induction of differentiation in mouse ESCs (Feldmann et al, 2020). Thus, it is emerging that a reorganization of long-range chromatin interactions represents an important part of IFN-mediated gene induction.

According to a motif analysis with HOMER software (Heinz et al, 2010), ISREs are present at 134,069 loci in the mouse genome. Our ChIP-seq analysis, however, yields a much lower number of 392 ISREs that actually had STAT1/2 bound. This large difference led us to characterize their chromatin environment as a determinant of STAT binding via a genome-wide correlation analysis. We find that a repressive chromatin conformation marked by H3K27me3 renders ISREs less accessible to STAT1/2 binding. In contrast, H3K4me1 and

H3K27ac and an open chromatin state detected by ATAC were associated with sites permissive for STAT1/2 binding. These results are in line with a previous study that compared histone modifications at 18 ISREs (Testoni et al, 2011). In the latter data set, six out of nine ISREs at activated promoters showed some enrichment for H3K4me1 before induction with IFNα. It is noted that H3K4me1 has been related to targeting the BAF (SWI/SNF) chromatin remodeler to chromatin, which interacts with STAT1$_{p701}$ and STAT2 via its BRG1 component. Accordingly, this histone modification could promote chromatin opening and subsequent STAT1/2 binding (Huang et al, 2002; Christova et al, 2007; Local et al, 2018).

In conclusion, our integrated multi-omics data set provides insight into the interplay between the IFNβ-mediated activation of ISGs, STAT binding, and chromatin features. It sheds lights on the mechanism that govern the cell type–specific IFNβ response as discussed above. This insight could be further exploited to modulate the IFN response during virus infection or therapeutic intervention in cancer (Hoffmann et al, 2015; Borden, 2019). Numerous so called "epigenetic drugs" that inhibit enzymes setting or removing histone acetylation and methylation are already used in anti-cancer therapy (Cheng et al, 2019; Mohammad et al, 2019). In the light of our study, the resulting perturbances of chromatin features are also likely to affect IFN response. For example, histone deacetylase (HDAC) inhibitors that result in the hyper-acetylation of histones and render chromatin more accessible could enhance STAT1/2 binding to otherwise occluded ISREs (Gorisch et al, 2005; Chen et al, 2022). At the same time, however, these drugs also affect the acetylation state of protein factors involved in IFN-mediated signaling like the acetylation and activity of the STAT1/2 complex itself (Tang et al, 2007). Accordingly, HDACs have been shown to both repress and enhance IFN response in a complex manner (Au-Yeung & Horvath, 2018; Lu et al, 2019). Thus, changing STAT1/2 binding patterns more specifically would require a targeted approach beyond global inhibition/activation of epigenetic modifiers like HDACs. This could be achieved, for example, by using more selective drugs (Cheng et al, 2019; Mohammad et al, 2019) or dCas9-mediated epigenetic editing of ISRE chromatin states at promoters and enhancers by targeted binding of activators that sets or removes H3K27ac, H3K4me1, or H3K27me3 (Li et al, 2020; Trojanowski et al, 2022). In this manner, ISG activation patterns and the cell type–specific antiviral response could be modulated.

# Materials and Methods

### Cell culture work and IFNβ treatment

Mouse 129/Ola ESCs, NPCs differentiated in vitro from ESCs, and MEFs were cultured at 37°C with 5% $CO_2$ and routinely checked for the absence of mycoplasma contaminations as described previously (Bibel et al, 2007; Teif et al, 2012; Mallm et al, 2020). IFNβ was prepared from a BHK cell line over-expressing IFNβ and grown with DMEM supplemented with 10% FCS, 1% L-glutamine, and 1% penicillin/streptomycin. After growing the cells in the same medium but with 2% FCS for 24 h, the IFNβ containing medium was passed through a 0.45-μm sterile filter and stored in aliquots at −80°C. The activity of the resulting IFNβ stock was determined against a commercial preparation of recombinant mouse IFNβ (Sigma-Aldrich) by treating an Mx2-luciferase reporter cell line (IEC-Mx2Luc-10) for 24 h (Schwerk et al, 2013). A stock concentration of 16.6 U/μl was calculated, and the dose-response curve of the reporter signal in IEC-Mx2Luc-10 versus IFNβ concentration from Schwerk et al was reproduced. For the experiments described here, cells were treated with IFNβ at a concentration of 500 U/ml for 1 or 6 h. At this concentration, a strong albeit not saturated IFNβ response was achieved in the cell types studied.

### Western blots

Western blot samples were prepared by collecting cells directly from the cell culture. The cells were transferred into 1.5-ml tubes, washed once with PBS, and counted. A 50 μl volume solution of pre-prepared RIPA buffer (150 mM NaCl, 1% NP40, 50 mM Tris–HCl, pH 8.0, 0.5% sodium deoxycholate, and 0.1% sodium dodecyl sulfate) was added per 0.5 million cells in suspension. The mixes were incubated for 60 min on ice, spun down at maximum speed at 4°C for 30 min. The supernatant was transferred to a fresh tube and stored at −20°C. Gels were blotted on LF PVDF membranes using the trans-blot turbo transfer system (Bio-Rad) and blocked with 5% BSA in Tris-buffered saline supplemented with 0.1% Tween 20 detergent (TBST) at room temperature for 1 h. The primary antibodies were diluted according to manufacturer's recommendations and incubated at 4°C overnight. On the following day, the membrane was washed three times with TBST buffer at room temperature for 5 min under agitation and incubated with secondary anti-HRP antibody (normally 1:5,000 diluted in 5% BSA) at room temperature for 1 h, washed three times with TBST, incubated with Clarity Western ECL substrate for 5 min, and imaged. The antibodies used are listed in Table S7.

### Bulk RNA-seq

Cells were seeded on a six-well plate. Two (ESCs and MEFs) or five (NPCs) days after plating, the cells were washed two times with PBS. Then, 500 μl LBP was added, and the cells were stored at −80°C. RNA was isolated with the NucleoSpin RNA kit (Macherey-Nagel) according to the manufacturer's protocol. The elution step was done two times with 30 μl RNase-free water within the same tube. Concentrations were measured by Qubit RNA HS Assay Kit, and the quality of RNA was analyzed on an Agilent 2200 high sensitivity RNA screen tape system. Removal of rRNAs from isolated samples of IFNβ-stimulated ESCs and MEFs was done following the protocol of the Ribo-Zero rRNA Removal Kit (Illumina). An input of 5 μg total RNA was used, and the depleted RNAs were eluted in 30 μl RNase-free water supplemented with 1 μl RiboLock RNase Inhibitor (40 U/μl). Concentrations were measured by Qubit RNA HS assay kit. For NPCs, RNA samples were treated with DNase at 37°C for 30 min and purified by ethanol precipitation. Concentrations were measured by Qubit RNA HS assay kit, and 750 ng of DNase-treated RNA was used for rRNA depletion by NEB Next rRNA Depletion Kit (human/mouse/rat). The depletion was performed based on the manufacturer's protocol. Samples were purified with RNA Clean XP beads (Beckman) with a 2.2× ratio and finally eluted in 8 μl nuclease-free water. Purified rRNA-depleted RNA samples of ESCs, MEFs, and NPCs were

used to prepare NGS libraries based on the NEB Next Ultra II directional RNA library preparation kit from Illumina. As default, 50 ng of rRNA-depleted RNA was used as input. For less concentrated samples, 10 ng were used. The RIN value of all samples was above seven, and therefore the mixes were incubated at 94°C for 15 min. Furthermore, a fivefold NEB Next adaptor dilution was used as default at the adaptor ligation step. For lower concentrated samples, a 25-fold dilution was used. All samples were dual-barcoded with unique i5 and i7 primers. For 50 ng samples, a total of nine cycles and for 10 ng samples 11 cycles were performed during the PCR enrichment of the adaptor ligation DNA step. Samples were measured with Qubit dsDNA HS assay kit, and the fragment size was determined with a Tapestation D5000. In total, four replicates of ESCs, two replicates of MEFs, and four replicates of NPCs treated with IFN$\beta$ for 0, 1, and 6 h, respectively, were acquired. The corresponding RNA-seq libraries were 50-bp single-end sequenced on a HiSeq 4000 System (Illumina) with at least 50 million reads per sample. Sequencing of RNA, and that of all other sequencing readouts, was done at the DKFZ Genomics and Proteomics Core Facility.

### ChIP-seq of STAT1p701 and STAT2

STAT1p701 and STAT2 ChIPs were performed with the ChIP enzymatic chromatin IP kit from Cell Signaling Technology according to the manufacturer's protocol. Around $4 \times 10^6$ cells per sample were used as input for the ChIPs. After formaldehyde fixation, chromatin fragmentation was done with the EpiShear Probe Sonicator (Active Motif) at 4°C with 50% amplitude and 6–10 "on" and "off" cycles of 30 s duration to yield an average fragment size of around 150 bp. The immunoprecipitation was conducted with 10 µg of chromatin in a total volume of 500 µl and addition of antibodies (Table S7). The sequencing libraries were prepared using the NEB Next Ultra II DNA library preparation kit for Illumina with 40 µl ChIP sample and added 10 µl 10 mM Tris–HCl, pH 8.0. For the input reaction, a 1:10 dilution was made and from this dilution 4 µg chromatin were used and filled up with 1 × 10 mM Tris–HCl, pH 8.0, to a total volume of 50 µl. Concentrations were measured by Qubit dsDNA HS assay kit, and fragment distribution was analyzed on a Tapestation D5000. The libraries were sequenced as described above for RNA-seq.

### ChIP-seq of histone modifications

ESCs were cultured in 150 mm dishes and treated with IFN$\beta$ for 0, 1, or 6 h. Media was removed and cells were detached with Accutase, washed with PBS supplemented with PMSF at 0.5 mM concentration, and crosslinked with 1% formaldehyde (1 ml 16% formaldehyde with 15 ml PBS) for 10 min at room temperature. 125 mM glycine was added to neutralize formaldehyde, and the cells were scratched from the plates on ice and collected in tubes. Afterward, the samples were washed three times with PBS/100 mM PMSF, and cell pellets were resuspended in 10 ml swelling buffer (25 mM HEPES, pH 7.8, 1 mM MgCl$_2$, 10 mM KCl, 0.1% NP-40, 1 mM DTT, and 0.5 mM PMSF). A 10 min incubation step on ice and a centrifugation step at 400$g$ for 5 min at 4°C were performed. $4 \times 10^7$ cells were resuspended in 100 µl MNase buffer and 40 U MNase was added. After an incubation step at 37°C

for 15 min, 100 µl of 10× sonication buffer and 800 µl water were added. Samples were incubated on ice for 5 min, transferred into 12 × 24-mm tubes, and sonicated for 15 min (burst 200; cycle 20%; intensity 8) on a Covaris sonicator. A centrifugation step was performed at 16,200$g$ and 4°C for 15 min. The supernatant was transferred into fresh tubes, and chromatin was snap frozen with liquid nitrogen and stored at –80°C. A quality check of reverse cross-linked samples was performed and yielded a fragment size of around 150 bp for the sheared chromatin. Pre-equilibrated 25 µl protein G beads were used per sample at room temperature for 10 min in sonication buffer (10 mM Tris, pH 8.0, 200 mM NaCl, 1 mM EDTA, 0.1% Na-deoxycholate, 0.5% n-lauroylsarcosine, and 0.5 mM PMSF). A sample precleaning step was performed by adding 25 µl protein G beads with 4 µg IgG antibody (rabbit or mouse) to chromatin and incubated, rotating at 4°C for 2 h. Beads were pelleted, and supernatant was transferred to fresh tubes. Antibodies were added to chromatin samples and incubated at 4°C for 2 h (Table S7). Then, 25 µl of pre-equilibrated beads were added to the samples and incubated rotating at 4°C overnight. The beads were washed by rotating at 4°C for 5 min with high-salt buffer (50 mM HEPES, pH 7.9, 500 mM NaCl, 1 mM EDTA, 1% Triton X-100, 0.1% Na-deoxycholate, 0.1% SDS, and 0.5 mM PMSF), lithium buffer (20 mM Tris, pH 8.0, 1 mM EDTA, 250 mM LiCl, 0.5% NP-40, 0.5% Na-deoxycholate, and 0.5 mM PMSF), and 2× with TE-buffer (10 mM Tris, pH 8.0, 1 mM EDTA). Each sample was eluted two times with 250 µl elution buffer (50 mM Tris, pH 8.0, 1 mM EDTA, 1% SDS, and 50 mM NaHCO2) at 37°C for 15 min on a shaker. Reverse cross-linking was performed by adding 20 µl 5 M NaCl and incubated at 65°C overnight. Subsequently, 10 µl EDTA (0.5 M), 0.5 µl RNase A (10 mg/ml), and 50 µl Tris (1 M, pH 6.8) were added and incubated at 37°C for 30 min. Then, 2 µl proteinase K (20 mg/ml) was added and incubated at 55°C for 2 h. An isopropanol precipitation was performed to purify the DNA. Resuspended samples were measured with Qubit dsDNA HS assay kit and the fragment size was determined on a D5000 Tapestation (Agilent). Libraries were sequenced as described above for RNA-seq. In ESCs, two replicates for H3K4me1, H3K4me3, and H3K27ac and three replicates for H3K9ac, H3K9me3, and H3K27me3 were sequenced. In MEFs, two replicates of all modifications were sequenced.

### Bulk ATAC-seq

ESCs were plated on six-well plates and treated for 0, 1, or 6 h with IFN$\beta$ at 500 U/ml. Cells were detached using Accutase, collected, and washed with MT-PBS buffer (4 mM NaH$_2$PO$_4$, 16 mM Na$_2$HPO$_4$ and 150 mM NaCl, pH 7.4). A total of 50,000 cells were transferred into fresh tubes and centrifuged by 800$g$ at 4°C for 5 min. For ESCs, the cell pellet was resuspended in 200 µl ATAC lysis buffer (10 mM TrisHCl, pH 7.4, 10 mM NaCl, 3 mM MgCl$_2$, and 0.1% NP-40), incubated at room temperature for 2 min and centrifuged at 800$g$ and 4°C for 5 min. The supernatant was discarded and pellets were resuspended in 20 µl ATAC reaction buffer containing 10 µl 2× transposase buffer and 2.5 µl Tn5 enzyme (Illumina). Samples were incubated at 37°C for 30 min. Reactions were stopped by adding 5 µl EDTA (100 mM) in Tris–HCl, pH 8.0, to a final concentration of 20 mM. For MEFs and NPCs, the cells were directly resuspended in 25 µl ATAC reaction buffer with digitonin (9.75 µl H$_2$O, 12.5 µl 2× transposase buffer

[Illumina], 0.5 μl 50× proteinase inhibitor, 2 μl Tn5 enzyme [Illumina], and 0.25 μl 1% digitonin) and incubated at 37°C for 30 min. The samples were purified with a MinElute PCR Purification Kit (QIAGEN) and eluted in 12 μl buffer. After PCR amplifications, sequencing libraries were purified with AMPure beads (Beckman). Concentration was measured with the Qubit dsDNA HS assay kit (Thermo Fisher Scientific) on a Qubit fluorometer, and size distribution of final library was checked on a D5000 Tapestation. Libraries were 50-bp paired-end sequenced on Illumina HiSeq 2000 or 4000 systems with at least 50 million reads per sample. Two replicates for ESCs and NPCs and four for MEFs were sequenced.

### Analysis of bulk sequencing data

For RNA-seq analysis, ribosomal RNAs were removed and raw reads were mapped with STAR (Dobin et al, 2013) to the mm10 mouse reference genome and normalized read counts (transcripts per kilobase million, TPMs) were computed with RSEM (Li & Dewey, 2011). The differential gene expression analysis between treated and untreated controls was performed using DESeq2 (Love et al, 2014), with $P$-value < 0.05 and log fold change >1.5. For the analysis of ChIP-seq and ATAC-seq data, reads were mapped with Bowtie2 (Langmead & Salzberg, 2012) to the mm10 mouse reference genome. Duplicates and reads annotated to blacklisted regions (Encode Project Consortium, 2012) and mitochondrial reads were removed. Quality control followed the Encode guidelines (https://www.encodeproject.org/data-standards/chip-seq/) and involved normalized strand coefficients and relative strand correlation values for each sample. Peak calling for STAT ChIP-seq was done with MACS2 (Zhang et al, 2008a). STAT1p701- and STAT2-binding sites were identified against the unstimulated controls for 1 and 6 h of IFNβ treatment from the ChIP-seq data with DiffBind (Ross-Innes et al, 2012) using the consensus peak list and thresholds of FDR <0.05 and fourfold enrichment. Sequence motifs enriched in STAT1, STAT2, and STAT1/2 peaks were identified using HOMER (Heinz et al, 2010). For the analysis of the STAT1/2 chromatin environment, STAT1/2 bound sites in ESCs and MEFs were expanded by 1 kb up- and downstream. The ChIP- and ATAC-seq signal in these regions was determined from the respective read counts after normalizing for library depth and fragment length and computing enrichments over histone H3 for histone modifications and IgG for STAT1/2. Replicates of the same samples and time points of IFNβ stimulation were averaged. The resulting count tables were used as input for the k-means clustering to characterize the chromatin environment at STAT1/2-binding sites.

### Single-cell RNA-seq and ATAC-seq

The scRNA-seq experiments were performed based on the standard protocol for the Chromium Single-Cell 3′ reagent kit v2 (10× Genomics). ESCs and MEFs were treated for 0, 1, or 6 h with IFNβ. The cDNA amplification was done by running 13 PCR cycles. The samples were eluted in 35 μl 10 mM Tris–HCl, pH 8.0. Concentrations of cDNA libraries were measured by Qubit dsDNA HS Assay Kit, and mean peak sizes of the samples were determined on a Tapestation D5000. Each of the final libraries were paired-end sequenced (26 bp and 74

bp) on one Illumina HiSeq 4000 lane following the manufacturing protocols. For scATAC-seq, ESCs and MEFs were treated with IFNβ for 0, 1 (only for MEFs), or 6 h, and libraries were prepared according to the Chromium Single-Cell ATAC v1.0 protocol (10× Genomics). Two (three for MEFs treated with IFNβ for 6 h) replicates per scATAC libraries were paired-end sequenced on an Illumina NovaSeq 6000 system, according to the manufacturer's protocol.

### Analysis of scRNA- and scATAC-seq data

Sample demultiplexing and barcode processing of scRNA-seq data were conducted with the Cell Ranger pipeline from 10× Genomics. For ESCs, quality filtering was conducted by selecting only cells within a certain percentage of mitochondrial reads (2.5% < accepted cells < 7.5%) and number of detected genes (2,000 < accepted cells < 6,500), yielding 1,332 cells for time point 0 h, 2,085 cells for 1 h, and 4,825 for 6 h of IFNβ stimulation. For MEFs, quality filtering was conducted by selecting only cells within a certain percentage of mitochondrial reads (0.5% < accepted cells < 7.5%) and number of detected genes (1,250 < accepted cells < 6,500), yielding 9,771 cells for time point 0 h, 10,186 cells for 1 h, and 7,579 for 6 h of IFNβ stimulation. Further analysis including UMAP embedding was done using the R package Seurat (Stuart et al, 2019). The scATAC-seq data were demultiplexed and aligned with Cell Ranger ATAC count (10× Genomics) using the provided mouse mm10 reference. Further processing of the data was conducted with ArchR (Granja et al, 2021). The cells were filtered using a minimal and maximal threshold for number of fragments ($10^{3.5}$ and $10^5$, respectively), a TSS ratio above 4 and a ratio of fragments in blacklisted genomic regions to all fragments below 0.0225 (ESCs) and 0.016 (MEFs). Cell numbers and quality measures of scATAC-seq data are provided in Table S6.

### Co-accessibility analysis

Regions of 2 kb size that were simultaneously open in the same cell within a 1 Mb window around STAT1/2 bound sites were identified from a co-accessibility analysis of the scATAC-seq data. Pearson correlation coefficients were calculated without aggregation of single cells for ESCs, epithelial-like and mesenchymal-like MEFs for each treatment condition based on our previously described RWire approach (Mallm et al, 2019) implemented into the ArchR framework (Granja et al, 2021) (https://github.com/RippleLab/RWire-IFN). For the co-accessibility analysis 2,700 cells per (sub-)type and treatment condition were selected that had a similar number of fragments per cell. The background co-accessibility signal was determined by randomly shuffling the accessibility values over cells and peaks (Mallm et al, 2019). As a threshold for the background correlation coefficient, the 99th percentile of shuffled background distribution was used, yielding $r$ = 0.07. Co-accessible links were further evaluated with respect to their $P$-value as computed by ArchR (significance level <0.01). As an additional parameter, the percentage of accessible cells was calculated as the average of cells that had site 1 or site 2 accessible with the rational that correlations computed from a low number of cells are less reliable. Only co-accessible links of STAT1/2 bound sites were considered for further analyses. We observed no correlation between the Pearson

correlation coefficients of co-accessible links and their percent accessible cells (Pearson $r$ = 0.02, $P$-value = 0.035) and pseudo-bulk number of fragments (Pearson $r$ = 0.03, $P$-value = 0.014).

## Data Availability

The data and computer code produced in this study are available from the following sources: All original sequencing and relevant processed data have been deposited under GSE160764 at Gene Expression Omnibus (https://www.ncbi.nlm.nih.gov/geo/). Software used for data analysis for the different sequencing readouts is listed in Table S8. The R-scripts for the co-accessibility analysis are available via Github at https://github.com/RippeLab/RWire-IFN.

## Supplementary Information

Supplementary information is available with this article.

## Acknowledgments

We thank Jorge Trojanowski, Sabrina Schumacher, and Katharina Bauer for discussions and help and Mario Köster and Hansjörg Hauser for providing the IFN*β* over-expression cell line. This work was supported by grants TRR179 (Z03) and RI1283/14-1 of the German Research Foundation (DFG) to K Rippe. We thank the DKFZ High Throughput Sequencing and the Omics IT and Data Management core facilities for support and services and the Central Animal Laboratory for preparation of mouse fibroblast cells. Additional data storage at SDS@hd was supported by the Ministry of Science, Research and the Arts Baden-Württemberg and the DFG through grants INST 35/1314-1 FUGG and INST 35/1503-1 FUGG.

### Author Contributions

M Muckenhuber: conceptualization, data curation, formal analysis, validation, investigation, visualization, and writing—original draft, review, and editing.
I Seufert: data curation, software, formal analysis, visualization, methodology, and writing—review and editing.
K Müller-Ott: data curation, formal analysis, and investigation.
J-P Mallm: investigation.
LC Klett: formal analysis.
C Knotz: investigation.
J Hechler: investigation.
N Kepper: data curation.
F Erdel: conceptualization, supervision, and writing—review and editing.
K Rippe: conceptualization, resources, supervision, funding acquisition, visualization, and writing—original draft, review, and editing.

### Conflict of Interest Statement

The authors declare that they have no conflict of interest.

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
