## [Reviewer comments · Life Science Alliance]

Life Science Alliance

Epigenetic signals that direct cell type specific interferon beta response in mouse cells

Markus Muckenhuber, Isabelle Seufert, Katharina Müller-Ott, Jan-Philipp Mallm, Lara Klett, Caroline Knotz, Jana Hechler, Nick Kepper, Fabian Erdel, and Karsten Rippe

DOI: <https://doi.org/10.26508/lsa.202201823>

Corresponding author(s): Karsten Rippe, German Cancer Research Center

Review Timeline:

Submission Date:	2022-11-11
Editorial Decision:	2022-12-13
Revision Received:	2023-01-14
Accepted:	2023-01-16

Transaction Report:

Please note that the manuscript was reviewed at Review Commons and these reports were taken into account in the decision-making process at Life Science Alliance.

General comments

We thank the editor and the expert reviewers for their insightful, constructive, and very helpful assessment of our study and their overall positive evaluation. We have included additional validations and analyses that address the various points raised and have thoroughly revised the whole manuscript as described below. The reviewer comments are highlighted in blue and have been renumbered to facility cross-references.

Reviewer 1

1. *“The study investigates parameters determining the response of genes to interferon-beta (IFN- β) in embryonic stem cells (ESC), ESC-derived neuronal progenitors (NPC) and mouse embryonic fibroblasts (MEF). Using bulk as well as scRNA-seq the authors demonstrate that the ESC response to IFN- β produces smaller transcriptome changes, that MEFs form distinct clusters based on their (IFN-induced) transcriptomes and that IFN-inducibility of some interferon-stimulated genes (ISG) varies between cell types. Consistent with this, genomic sites associating with pSTAT1 and STAT2 differ between ESC and MEFs. As some ISG cannot be directly assigned to an IFN-response element in their promoters, the authors use co-accessibility of chromatin based on scATAC-seq data to link IFN-induced gene expression with the activity of distal enhancer elements. They conclude that a subset of genes is indeed driven by distal enhancers in absence of promoter proximal elements and that this mechanism operates in clustered ISG. Further data are shown to define chromatin marks associated with responsiveness to IFN. Somewhat unsurprisingly, the authors find a correlation between active histone marks, STAT binding and gene inducibility.”*

We are grateful to the reviewer for the positive evaluation of our work and have addressed the specific major and minor comments as described in the following.

Major comments

2. *“The study contains interesting findings about the responsiveness of potential ISG in different cell types. Unfortunately, the dataset with NPC is very limited. It would have been of interest to use this cell type in the entire set of experiments because it stands in a direct relationship to ESC and conclusions about differentiation-related changes in the IFN response could have been drawn. As such, the data comparing the cell types remain descriptive.”*

Our study starts with the comparison of ESCs, NPCs and MEFs from the same mouse strain with respect to IFN β induced gene expression to identify epigenetically driven differences. While NPCs were generated by direct *in vitro* differentiation of ESCs, MEFs were obtained from the same 129/Ola mouse strain. Thus, all three cell types are genetically identical, and we considered all types of pairwise comparisons between them as informative on the

epigenetic/cell type specific contribution to IFN β response. The biggest difference in the ISG transcriptome profiles was observed in the ESC vs MEF comparison. Following the suggestion of the reviewer (also see point #19), we have now moved the NPC RNA-seq data from main **Fig. 1** to **Supplementary Fig. 1** but kept them in the manuscript. Subsequently, we focused on the ESC-MEF comparison. We believe that the ESC-NPC and MEF-NPC comparison of the transcriptome (and providing the associated sequencing data) is valuable. It can be used with the large body of chromatin feature data that have been previously acquired for NPCs and include, for example our nucleosome position maps (Teif et al, 2012) as well as histone modification ChIP-seq data (Molitor et al, 2017).

3. “The co-accessibility approach is very interesting and the data about the accessibility links in clustered ISG correspond very well with some recent observations (see specific comments).”

We thank the reviewer for appreciating the value of our approach of calculating co-accessibility from scATAC-seq data to link regulatory elements to target genes and have addressed the specific comments as described below at points 10-13.

4. With regard to the clarity and structure of the manuscript it fails to generate a consistent narrative with regard to its aims. It remains unclear whether it mainly aims at generating fundamental insight into the chromatin structure -IFN response relationship or whether it mainly aims at demonstrating cell type-specific differences. The authors should attempt to clarify how the cell type comparison increases knowledge about IFN responses over a study performed with one cell type alone.

We have revised the introduction to clarify our aims and the associated structure of the manuscript. In our opinion, chromatin features and the differences in the cell type-specific response to IFN treatment are closely linked as the epigenetic makeup of a cell determines its gene expression programs and with it the cell type. We consider the specific comparison of ESCs vs. MEFs as an exemplary case to reveal more general and fundamental features of chromatin structure (e.g., open chromatin, H3K4me1, H3K27ac at promoters and enhancers) that are relevant for directing IFN response also in other cell types.

Specific comments

5. “Figure S1 B: It would be helpful to annotate the same genes as in fig. 1B”

We agree and have updated the data accordingly as **Supplementary Fig. 1D**. The same twelve genes were now highlighted in the plots.

6. *“Figure S1D: Please explain what information S1D provides in addition to that in Fig. 1D.”*

We have revised **Fig. 1** and have deleted the two UMAPs that show the cluster assignment at the bottom of **Fig. 1D**. Labels were added to the remaining UMAPs to indicate the numbering of clusters. **Supplementary Fig. 1F** (previously Supplementary Fig. 1D) was kept. It is needed to illustrate the assignment of MEF clusters to the epithelial to mesenchymal transition (EMT) based on the varying contribution of EMT-associated principal component (PC) 2 from PCA to the clusters.

7. *“Figure 3: Promoters/enhancers associating with STAT1 and STAT2 might have adjacent ISRE and GAS sequences. This cannot be resolved by ChIP-seq.”*

It is unclear to us what this comment aims at. In **Fig. 3C**, we have conducted a TF motif analysis in ESCs and MEFs for three different types of STAT peaks (only STAT1_{p701}, only STAT2 and peaks that carry both STAT1_{p701} and STAT2). The results showed enrichment of different STAT/IRF family motifs with the most frequent ones being also listed in **Supplementary Fig. 3A**. The quantification adjacent to the enrichment plots in **Fig. 3C** describes the composition of motifs found in the peaks according to four motif categories: “STAT family”, “IRF family”, “STAT & IRF family” and “Other”. Thus, the simultaneous occurrence of ISRE and GAS sequences in ChIP-seq derived peaks was identified and is represented by the peaks that fall into the STAT & IRF category. For example, in STAT1_{p701} peaks in ESCs, 11.5% of peaks belong to the STAT & IRF category and therefore contain both an IRF and STAT motif. In addition, **Supplementary Data Set 2** provides genomic locations of all peaks for further in-depth analysis of their sequences.

8. *“Fig. 3D: please provide information whether the annotation of STAT peaks to ISG refers to the cell type-specific ISG according to the author's own RNA-seq data or to more general ISG assignments.”*

We believe that the reviewer refers to **Supplementary Fig. 3D** since the main **Fig. 3D** does not differentiate between ISG and non-ISG promoters. It is now stated in the main text and in the legend to **Supplementary Fig. 3D** that the annotation as an ISG promoter is based on the ISGs identified in our study (**Fig. 1B**, **Supplementary Data Set 1**).

9. *“The finding that genes associated with STAT1 alone are expressed in absence of IFN and poorly inducible is surprising. Did the authors specifically look at genes generally thought to be regulated by STAT1 homodimers such as Irf1, Irf8 and Socs3?”*

We have realized that this information was not easily accessible from the manuscript. Accordingly, we have now added the data from **Supplementary Fig. 3** to **Supplementary**

Data Set 2. One can now directly look up genes with STAT peaks at the promoter and the STAT binding status at ISG promoters with ISGs being listed in **Supplementary Data Set 1**. For the three genes mentioned (*Irf1*, *Irf8* and *Socs3*), this yielded the following results: *Irf1* and *Socs3* were ISGs in ESCs and MEFs (**Supplementary Data Set 1, Fig. R1**). Their promoters were bound by STAT1 and STAT2 and therefore assigned to the STAT1/2 category. According to our gene expression analysis, *Irf8* was not an ISG and bound by neither STAT1 nor STAT2. Furthermore, the chromatin environment at the *Irf8* promoter exhibited low levels of H3K4me1 and lacked an ATAC signal for open chromatin. In the literature *Irf8* is described as being expressed primarily in myeloid cells like macrophages and dendritic cells and its activation by IFN γ or lipopolysaccharide (LPS) (Marquis et al, 2011) is driven by STAT1 homodimers (Ivashkiv, 2018).

Fig. R1. Normalized gene expression levels of *Irf1*, *Irf8* and *Socs3* from bulk RNA-seq in ESCs (top) and MEFs (bottom). Gene expression is given as transcripts per kilobase million (TPM).

10. Figure 4D, left: is the pseudo bulk display limited to the very small fraction of cells that express *Uba7*? Otherwise, I don't see how a meaningful conclusion could be drawn. The figure would be more convincing if tracks showing induced binding of STAT1/2 to the intronic enhancer were included. The placement of the greyscale map for the correlation coefficients is not well chosen as it refers to a different part of the figure as the symbols and color codes for the expression data. The authors should also provide an explanation what cell fraction is meant on the y-axis.

We are grateful for the comment and the instructive suggestions to improve **Fig. 4**. We agree that additional information on STAT1/2 binding events should be provided for the exemplary regions. We decided against including the tracks for STAT1/2 ChIP-seq data since **Fig. 4D, E**

are already quite complex. Instead, we have added STAT1/2 peak IDs and additional information on STAT1/2 binding events to the **Fig. 4D, E** legend and **Supplementary Data Set 3**. Moreover, we have adapted the y-axes and legends of the loop tracks as suggested by the reviewer.

The pseudo-bulk is, by definition, the aggregation of all the data sets from individual cells. We also would like to emphasize that the co-accessibility analysis cannot be limited to only the fraction of cells that express a given ISG. The stochastic switching between on- and off states (both at the level of chromatin accessibility and gene expression) is the crucial information for computing the co-accessibility correlations. Selecting only a fraction of cells that express a given gene would confound this analysis. It is noted that a high fraction of cells with “co-inaccessibility” could make the result less reliable, since inaccessibility might not only derive from true biological inaccessibility but also from technical “drop-outs”. This is accounted for by using a sufficiently high number of cells to compute the correlation coefficient and assessing its significance statistically and against a background model (see also response to #35 below). To clarify our approach for identifying co-accessible links the corresponding section in Materials & Methods has been expanded.

11. Fig. 4E: see comments to 4D. There is no explanation why the co-accessibility pattern differs between epithel-like and mesenchymal-like MEFs, but the transcriptional output of the Ly6 genes is largely the same.

The reviewer addresses the interesting point that one would expect similar gene expression regulatory mechanisms for Ly6 in epithelial- and mesenchymal-like MEFs if the transcriptional output is the same. However, we also observe differences in the pseudo-bulk accessibility profiles of the two MEF subtypes. While IFN β treatment induces a clear overall increase of accessibility at the locus, the changes vary considerably between individual sites (see marked regions in **Fig. 4E**). This suggests to us that there is indeed a difference in the underlying regulatory networks that could arise, for example, from the differential activity of loop mediating interactions that affect combinatorial enhancer-promoter interactions. Alternatively, the data sparsity of scATAC-seq data might also lead to missing regulatory interactions present in a MEF subtype. We have now briefly addressed this point in the Results part of the main text.

12. The notion of crossregulation of clustered ISG is in agreement with two recent publications (<https://doi.org/10.1038/s41467-021-26861-0>; DOI: 10.1016/j.isci.2022.103840).

We agree with the reviewer and now reference the above-mentioned publications in the discussion. It supports our view that this emerging topic is an important aspect of interferon

stimulated gene expression. We also have added information on distal ISG regulation by promoters and/or enhancers in **Supplementary Fig. 4E**.

13. *“Fig. 4G: The different categories are shown as though they were mutually exclusive. Are there no genes with promoter binding sites that show co-accessibility with distal enhancers?”*

This point is well taken. It is now stated in the text that the ISG regulation categories are not mutually exclusive. We assigned ISGs to the ISG regulation categories as follows: (i) ISGs with STAT1/2 ChIP peak at promoter were assigned to “STAT1/2 binding at promoter” category (independent of the presence of additional links to distal sites). (ii) ISGs without STAT1/2 at promoter that showed a co-accessible link to a distal STAT1/2 peak after IFN β induction were classified as “Gained link to distal STAT1/2”. (iii) ISGs without STAT1/2 at promoter that showed a co-accessible link to a distal STAT1/2 peak only before IFN β induction were assigned to “Lost link to distal STAT1/2” category. (iv) The fourth category “Other” comprises ISGs that neither have STAT1/2 at the promoter nor show a co-accessibility link to a distal STAT1/2 site. We clarified the assignment of ISGs to regulation categories in the Figure legend and added information on ISGs with multiple regulatory STAT1/2 processes in **Supplementary Figure 4D**.

14. *Fig. 5A: In the ESC the signal-to-noise ratio of the STAT2 track (Ifi27, 6h IFN) doesn't suggest high quality data. Likewise, the 'enhancer' mark H3K4me1 is found throughout the gene body.*

We thank the reviewer for this comment. When double-checking the IGV tracks of the figure we found an error in **Fig. 5A**. During figure assembly the 6 h IFN β H3K4me1 signal was accidentally duplicated and misplaced upon the STAT2 track. This error has been corrected now (**Fig. R2**).

Fig. R2. Browser tracks of STAT2 and H3K4me1 at Ifi27 promoter in ESCs at the indicated timepoints of IFN β treatment (0 h, black; 1 h, red; 6 h, blue). Tracks at the top show the previous Fig. 5A while the corrected STAT2 6h tracks is shown at the bottom.

As pointed out by the reviewer “H3K4me1” is a marker for the active enhancer state. However, it is also well established that H3K4me1 is found at the flanking regions of active promoters next to H3K4me3 or at the center of poised promoters together with H3K4me3 and H3K27me3 and can display rather broad distributions within the gene body (Bae & Lesch, 2020; Barski et al, 2007; Cheng et al, 2014; Ernst et al, 2011; Molitor et al, 2017). It is noted that a large fraction of enhancers is intronic and thus would have H3K4me1 if active. Furthermore, the ENCODE guidelines for histone ChIP-seq (<https://www.encodeproject.org/chip-seq/histone/>) consider H3K4me1 as a mark that requires parameters for calling rather broad peaks, in contrast to H3K4me2 and H3K4me3 (Marinov et al, 2014). Thus, our data on the genomic distribution of H3K4me1 are fully consistent with those reported previously.

15. The data for MEF do not include histone marks in the IFN-induced state. How is it possible to derive relationships with enhancers activated by IFN treatment with this data set?

The relationships between enhancers and IFN-induced ISG promoters described in **Fig. 4** and **Supplementary Fig. 4** and **Supplementary Data Set 3** were based on the scATAC-seq data. As described in the literature and in our own work (e.g., (Mallm et al, 2019) and reference therein) the open chromatin state identified by ATAC is in general equivalent to the combination of H3K27ac and H3K4me1 for identifying the formation of an active enhancer state.

16. Fig. 5C: This comparison should not be called or graphically represented as a transition as this term suggests that one cell type was directly derived from the other. There could be many intermediate states before arriving at the fibroblast epigenome from an ESC starting point.

This point is well taken. The figure displays chromatin state transitions between corresponding genomic positions as defined by STAT binding in the two cell types. We have now stated explicitly in the figure legend that the linkage does not imply that there would be a direct differentiation path from ESCs to MEFs.

Summary

17. Knowledge about the role of chromatin rearrangements in ISG activation is currently emerging. To my opinion, the part of the manuscript describing co-accessibility is the greatest scientific advance as it suggests that some ISG are regulated from distal enhancers and that enhancers or proximal promoter elements may be used to cross-regulate clustered ISG. Therefore, the respective parts of the manuscript should be improved with a better explanation of the approach and the changes to the figures as suggested.

We are glad to learn that the reviewer appreciates the value of our co-accessibility study. As described in our response to points #10-13 above, we have improved this part and expanded

the explanation of the co-accessibility analysis in the Materials & Methods, Figure legends and main text. The scripts for the co-regulation analysis are provided via Github at <https://github.com/RippeLab/RWire-IFN>.

18. The manuscript would further benefit from adding ChIP-seq data analyzing H3 marks in IFN treated cells.

We agree that histone ChIP-seq data of IFN β treated MEFs in addition to the ChIP-seq data of IFN β (un)treated ESCs and the ChIP-seq of untreated MEFs would also be interesting. However, we do not consider these data essential for the points that we make in our and thus refrain from further extending the already quite extensive data set provided with our manuscript (**Supplementary Table 1**). We also would like to point out that the amount of genome wide data provided in our study compares favorable to other studies on the relation of chromatin features and IFN mediate gene activation. Other studies, for example, only use selected ISG promoters to characterize chromatin states by ChIP with PCR readout.

19. My final suggestion is to either add more data with NPC or to omit them entirely. There is no benefit from including the currently available data set.

See comment to point #2 above.

Reviewer 2

Summary

20. Muckenhuber et al present an in-depth analysis of the chromatin status and STAT1/2 binding in three different mouse cell types before and after their treatment with type 1 interferon. It is the aim of the study to better understand cell type-specific differences in the expression of interferon-induced genes. They find that embryonic stem (ES) cells and embryonic fibroblasts (EF) are most divergent when it comes to interferon responses. Fewer genes are induced by interferon in ES cells compared to EF cells, and the authors attribute the differences to a chromatin state that is less accessible to transcription factor binding.

Major comments

This work is impressive in both depth and breadth of the analyses. The paper is well written and strikes a balanced tone. The authors make very good use of the massive amount of data they gathered and present it generally well. This said, as someone who is not intimately familiar with the representation of RNA-seq data, some plots are beyond me, such as the UMAP diagrams in Fig. 4, or the density curves of gene expression shown in Suppl. Fig. 2B. This is a general problem with ever more technically specialized literature, and one that limits accessibility to the readership. Additional explanations in the supplementary materials are required.

We thank the reviewer for appreciating the depth and value of our study. Indeed, it is challenging to present the large amount of complex data in the figures in a manner that allows a straightforward access to the relevant information for experts from various fields. We have expanded the legends to **Fig. 1**, **Fig. 4** and all supplementary figures to address the reviewer's request for additional explanations of the visualization of the data and have revised the text to facilitate the understanding of some of the relevant technical aspects.

21. The authors' analysis of cell type-specific gene expression and STAT1/2 binding finds 33 ES cell-specific genes and several hundred ES cell-specific ChIP-seq peaks. Whether there is a discernible link between these populations, i.e. whether ES-specific STAT binding events mainly occur at ES-specific ISGs, does not seem to have been resolved. As the data is available and interesting, I think this would be a worthwhile addition.

We are grateful for this suggestion. The complete information about the relation of STAT binding at ISG promoter is provided in **Supplementary Data Set 3**, worksheets "ISG_TSS_w_STAT_ESC" and "ISG_TSS_w_STAT_MEF". We intersected the 33 ESC specific ISGs (**Supplementary Data Set 1**) and the ISGs being bound by STAT1/2 in ESCs (**Supplementary Data Set 3**). Out of 33 ISGs we identified three genes (*Shisa5*, *Trim56*, *Ifi27*) as being directly bound by STAT1/2 at the promoter. Thus, the cell type specific ISGs in ESCs seem to be regulated mainly via non-promoter binding of STAT1/2. This is in line with the data shown in **Fig. 4** and **Fig. 5** that point to cell type specific enhancers as a crucial determinant

of a cell type specific IFN response. This strengthens the value of the approach described in the context of **Fig. 4** to link STAT1/2 binding at non-promoter sites to ISGs via the analysis of co-accessibility correlations between loci computed from scATAC-seq data.

22. In conclusion, the paper provides a wealth of insights about the correlation of chromatin modifications, the access of STAT proteins and the resulting transcription responses.

We thank the reviewer for his/her favorable assessment of our work.

Minor comments

23. Fig 1C is said to show the overlap of ISGs found after 1 and 6h interferon treatment. However, this is not apparent from the description provided.

The number of ISGs shown in **Fig. 1C** indeed represents the combined list of ISGs identified after 1 h or 6 h of treatment. This has been clarified in the main text and in the figure legend.

24. On page 7, the authors give numbers for the abundance of STAT- and IRF-family sequence motifs in the mouse genome but a justification/explanation for their numbers is missing.

It is now stated that the number of motifs referred to in the manuscript was extracted from the HOMER database. It provides the annotation of all known motifs in various reference genomes like the mouse mm10. From this file, we extracted and counted all motifs highlighted in **Supplementary Fig. 3**.

25. On page 15, the authors mention "closed" genomic distance. This must be a typo.

The typo "closed" instead of "closest" has been corrected.

Reviewer 2 (Significance (Required)):

Significance

26. The paper adds to a growing list of studies that explore the link between interferon-induced transcription, the chromatin landscape and DNA recruitment of STAT proteins. It differs in some respects from previous works (for example on the role of interferon-independent STAT2 chromatin binding, or the importance of STAT2-IRF9 complexes for interferon signalling), but generally confirms the current thinking in the field, namely that ISGF3 is the main driver of type 1 interferon responses, and that its activity is not limited to gene proximal promoter elements. The finding that pre-stimulation chromatin features are another determinant of transcription is less well documented in the interferon context but not unexpected. Regarding the latter point, it might be helpful to briefly compare interferon signalling with other signalling pathways in ES cells regarding the consequences of chromatin accessibility for gene expression.

We appreciate the suggestion to compare interferon signaling to other signaling pathways in ESCs and have briefly addressed this point in the discussion. It is well established, that ESC chromatin is in a more plastic “hyperdynamic” state with distinct differences in epigenetic modification patterns and chromatin accessibility as compared to differentiated cells (Lim & Meshorer, 2021). The pluripotency and self-renewal capacity of ESCs is linked to this chromatin state as well as the specific activity of several signaling pathways and cytokine response (Gordeeva, 2019; Kristensen et al, 2005). That includes an inverse correlation between the response IFN-I system and the maintenance of pluripotency (Eggenberger et al, 2019).

27. Finally, I think the authors are right to conclude that chromatin modification is a determinant of cell type-specific differences in interferon signalling. But how important is it? The data in Fig. 2B and Suppl. Fig. 2B are relevant in this regard. They show constitutive (interferon-independent) gene expression and protein content for ES and EF cells of the crucial transcription regulators STAT1 and STAT2. In terms of transcription, the two cell types don't differ. Protein contents, in contrast, differ strongly. Based on the western blot data of Fig. 2B, there is at least 25 times more STAT1 and STAT2 protein in EF cells compared to ES cells. This suggests that post-transcriptional differences play a major role. This consideration, in my view, should be pointed out in the discussion.

The summary of the reviewer is accurate and describes our data very well. We agree with the reviewer that the differences could be partially explained by post-translational modifications and have now better covered this aspect at the beginning of the Discussion section. At the same time, we would like to note that our ChIP-seq data for STAT1/2 yields similar numbers of STAT peaks in ESCs and MEFs. Thus, despite the different protein levels, the number of active STAT complexes bound to chromatin was similar. Together with the other data we have acquired and analyzed in the manuscript we conclude that also chromatin states represent an additional layer for directing the IFN β response.

28. This paper will be of interest to researchers working on mechanisms of inducible and cell type-specific gene transcription, in particular the interferon community, of which I am a member.

We thank the reviewer for his/her constructive comments and suggestions on our manuscript.

Reviewer 3 (Evidence, reproducibility and clarity (Required)):

Summary

“Authors aimed to uncover the linkage between transcription induction of ISGs and STAT1/2 DNA binding before and after treatment with IFN-beta. Further, authors identify most important histone H3 modification as well as sites of open chromatin by ATAC. Results were presented mainly for two cell types, namely embryonic stem cells and embryonic fibroblasts. Neural progenitor cells were only analyzed for ISG induction. Main conclusions: i. Cell type specific differences in ISG expression levels were observed upon IFN-beta stimulation between ESCs and MEFs/NPCs; this includes the set of genes induced and expression strength; ii. Binding of both factors, STAT1 and STAT2, at promoter sites correlates partly (41% for ESCs and 49% for MEFs) with ISG activation; iii. 25% of ISGs without STAT1/2 promoter binding were linked to a distal STAT1/2 binding event; iv. Pre-existing active chromatin marks (H3K4me1 and H3K27ac) correlate with STAT1/2 binding while H3K27me3 modification impeded this interaction.”

Major comments:

29. Chapter: IFN β induces anti-viral gene expression programs in all three cell types

“By intersecting the three individual ISG sets, we obtained 143 common ISGs while 33 (ESC), 17 (NPC) and 221 (MEF) ISGs were cell type specific (Fig. 1C, Supplementary Data Set 1).” Authors use for all three cell types 500 U/ml IFN-beta. Is this concentration for all cell types at the saturating level? Wang et al. 2014 (J Biol Chem. 2014 Sep 5;289(36):25186-98) used up to 5000 U/ml IFN-beta to achieve protection against LACV-induced cell death. Can the authors exclude dose-dependent differences (especially in the ESCs) rather than cell type specific differences? Authors should perform dose-dependent RT-PCR analysis of selected genes in ESCs (common versus MEF-specific genes).

At the beginning of our study, we tested concentrations and treatment times with IFN β for gene expression induction by RT-PCR of selected genes (IRF1, IRF3, IRF7 and ISG15). We selected 500 U/ml IFN β and 1 h and 6 h time points as experimental conditions that yielded relatively high induction in ESCs in a regime that was similar to conditions used in other studies for strong ISG induction in various cell types (Bolivar et al, 2018; Burke et al, 2011; Platanitis et al, 2019). In line with these considerations, our scRNA-seq data show a homogenous response after 6 h of IFN β treatment in both ESCs and MEFs (**Fig. 1D-F**) and a similar number of STAT1/2 peaks (in ESCs 208 vs in MEFs 276) (**Fig. 3B**). Thus, we consider our conditions appropriate for comparing the two cell types. We do not claim that the chosen IFN β concentration and 1 h or 6 h time points would correspond to conditions for maximal or saturating ISG expression in neither MEFs nor ESCs. This aspect is now mentioned at the beginning at the results and discussion section, and we point to the contribution of IFN β concentration or the treatment time on gene expression response and STAT binding. Furthermore, it is noted that determining the IFN β concentration for full protection against LACV infection after 24 h IFN β as done in the Wang et al. study (Wang et al, 2014) is on a different time scale and only indirectly related to a concentration of saturating ISG expression. As stated in the later paper the assay conditions were changed to a 10-fold higher virus particle

number as ESCs showed a very low infection rate already in the absence of IFN β . This suggests the activity of other interferon independent protection mechanisms (Maillard et al, 2013; Poirier et al, 2021), which would be a potential confounding factor when using a virus infection protection assay as a proxy for ISG induction in ESCs.

30. *“Chapter: ISG expression varies between cell types in response strength and specificity
“Next, we compared the transcriptional response to IFN β in the three cell types in further detail.” Detailed results are presented only for ESCs and MEFs. Authors should present corresponding data for NPCs to claim that detailed analysis for the three cell types are part of the manuscript. “*

This issue has also been raised by Reviewer 1. We have addressed it as described above at point #2.

31. *“... while for key transcription factors Stat1, Stat2 and Irf9 no differences were identified (Supplementary Fig. 2B). A western blot with STAT1 and STAT2 antibodies showed that STAT1 and STAT2 proteins were present at lower levels in ESCs before and after IFN β induction as compared to MEFs (Fig. 2B).’ STAT1/2 protein levels are much higher in MEFs compared to ESCs at unstimulated conditions (taking into account of much lower GAPDH levels in MEFs). However, authors identify no significant differences at RNA expression level (SupFig 2B) before IFN stimulation. Further, STAT1 RNA levels increases dramatically upon IFN-beta stimulation (6h, Fig. 2A) but no change in protein level is obvious (Fig. 2B). This paragraph is contradictory and authors not comment on differences between RNA levels and amount of STAT proteins.*

The reviewer points out a lack of correlation between RNA and protein levels for STAT1 and STAT2 when comparing ESCs and MEFs. It is well established that the gene specific correlation between RNA and protein levels is weak with RNA/protein ratio differences of three orders of magnitude, most likely due to differences in the rates of RNA translation and degradation (Edfors et al, 2016). However, large variations of this ratio for the same protein between different cell types are not common, which is indeed unusual finding from our STAT data. A number of reports have shown that STAT1 and STAT 2 protein degradation is regulated in a complex manner and dependent on its post-translational modifications (Kok et al, 2020; Lee et al, 2020). Thus, we conclude that the underlying circuitry that determines the relation between RNA transcript abundance and STAT1 and STAT2 protein levels is significantly different between ESCs and MEFs and leads to a reduction of STAT protein levels in ESCs. This is now mentioned in the context of **Fig. 2B** and **Supplementary Fig. 2B** and in the discussion.

32. *“In Fig. 2C authors show normalized gene expression levels of three cell type-specific ISGs in ESCs and NPCs/MEFs. Here it is not explicit mentioned but results may come from bulk*

RNA-seq analysis. Validation by independent RT-PCR analysis would be more appropriate for this main statement.”

We have chosen RNA-seq and not RT-PCR for our study to be able to measure differential gene expression in response to IFN β in different cell types in a genome-wide manner and not only for pre-selected genes. The quantification of gene expression is given as normalized counts (TPM, transcripts per million) from bulk RNA-seq, which is now also stated in the legend to Fig. 2C. Our protocol for measuring differential gene expression by bulk RNA-seq with 2-4 replicates per condition, high sequencing depth and analysis with the DESeq2 software is well established as a reliable approach. Furthermore, it is noted that our bulk RNA-seq data are fully consistent with the pseudo-bulk data from the scRNA-seq analysis, which counts uniquely barcoded single transcripts and is thus independent of PCR amplification artefacts. Thus, we are convinced that our RNA-seq data are of high quality. Comparison of RNA-seq pipelines vs. RT-PCR have been made numerous times (e.g., (Liu et al, 2022) and references therein), and we do not see the need to repeat this type of benchmarking and validation. That being said, we have conducted RT-PCR for selected genes when initially selecting the IFN β concentration and treatment duration. For example, the comparison of RNA-seq vs RT-PCR in ESCs at 500 U/ml IFN β for 1 h treatment yielded expression changes of 4.1 vs. 4.0-fold (*Irf1*), 1.0 vs. 1.2-fold (*Irf3*), 80.8 vs. 27-fold (*Irf7*) and 66.5 vs. 7.3-fold (*Isg15*). Thus, in these initial tests qualitatively similar results between RT-PCR and RNA-seq analysis were obtained but absolute fold-changes were somewhat different for *Irf7* and *Isg15*.

33. *“Chapter: ISG activation can be partly assigned to STAT promoter binding*

In Tab. S1 two time points of IFN-beta stimulation are indicated for ChIP-seq of STAT1p701 and STAT2. In the corresponding chapter no discrimination between early and rather late (regarding STAT DNA binding) time points are given. Authors should indicate if conclusions of STAT binding are made from both time points and how DNA binding changes over time (promoter versus non-promoter sites).”

We have now clarified in the text and the figure legends that we did not differentiate between STAT binding sites at 1 h and 6 h IFN β treatment for the analysis of STAT binding presented in **Fig. 3** and **Fig. S3**. When analyzing the time points separately, differences were minor and we identified identical motifs for STAT1, STAT1/2 and STAT2 groups. The full data set that dissects STAT1 and STAT2 binding at the different time points is provided in **Supplementary Data Set 3**.

34. *“Concrete examples of STAT1/STAT2 binding and ISG activation (results from Fig. 1) in the different cell types is missing. Authors should combine both results and give concrete examples. E.g. for common genes and ISG that are expressed only in MEFs or ESCs.”*

We provide exemplary tracks for all readouts used in our study in **Fig. 5A**. Additional STAT bindings tracks are shown in **Fig. 3A**. For the complete results of our analysis, we refer to the Supplementary Data Sets 1-3 and the bed files provided on GEO (GSE160764).

35. *“Chapter: STAT1/2 enhancers are predicted from co-accessibility analysis*

‘As an exemplary result, induction of the Uba7 ISG by STAT1/2 binding to a putative distal enhancer in ESCs is depicted in Fig. 4D. The IFN β -induced co-accessible link between the STAT1/2 bound enhancer candidate and the Uba7 promoter was associated with an increase in Uba7 expression in scRNA-seq.’ Uba7 expression can be detected in ca. < 10% of cells (Fig. 4D). Is this example significant to assume a co-accessible link between the STAT1/2 binding and Uba7 promoter activation?

Gene expression and accessibility are stochastic and, for technical reasons, only a fraction of the signal present is detected in single cell sequencing-based analyses. This leads to a rather sparse data matrix in a given single cell for both readouts. Thus, expression of a given gene in only a relatively small fraction of single cells is inherent to the method. For Uba7, expression is detected in 4.4 % of ESCs treated with IFN β for 6h, which represents a clear gene expression signal for a single cell data set. As now described in the expanded Materials & Methods section, we apply three criteria that need to be fulfilled to consider a co-accessible link significant: (i) The correlation coefficient is above the background co-accessibility threshold of 0.07. (ii) The p-value of the correlation coefficient as computed with the ArchR co-accessibility framework is <0.01. (iii) The percentage/number of accessible cells is calculated as the average of cells with accessible site 1 and cells with accessible site 2 (2.2% or 60 cells for Uba7). It is used to assess whether the number of cells used in the analysis is in the range at which cell (sub)types can be reliably identified within a single cell data set. Since these three criteria are fulfilled for the Uba7 example, we consider the co-accessible link significant that is shown in the **Fig. 4D**.

36. *Does the STAT1/2 binding indicated by the green vertical bar based on 1h or 6h ChIPseq results? Expression is high only after 6h; correlates this with STAT1/2 binding at putative enhancer region at late time point?*

In **Fig. 4D, E**, STAT1/2 binding sites marked by green vertical bars include differential STAT1/2 peaks after 1h and/or 6h of IFN β treatment. We added STAT1/2 peak IDs in the figure legend together with further information on STAT binding **Supplementary Data Set 3** to clarify this point. The STAT1/2 bound site 371 in the *Uba7* locus is bound after 1h of IFN β treatment in ESCs and accessibility of this site 371 further increases after 6h of IFN β treatment at which timepoint also induced gene expression is detected. This delayed response to STAT1/2 enhancer binding might reflect the need for additional steps to induce *Uba7* transcription, e.g., binding of additional factors and/or reorganization of chromatin folding at the locus.

37. Discussion: *"Thus, an activity of unphosphorylated STAT2-IRF9 for basal gene expression of ISGs as reported in (Blaszczyk et al, 2015) was not apparent in the STAT2 binding maps recorded here."*

Platanitis et al. (2019, A molecular switch from STAT2-IRF9 to ISGF3 underlies interferon-induced gene transcription. Nat Commun 10, 2921 (2019) conclude from their work that "... that the signal-independent formation of STAT2-IRF9 complexes, previously considered as noncanonical, is an integral component of ISG regulation. The change from STAT2-IRF9 to ISGF3 functions as a molecular switch between resting and active states for many ISGs." Platanitis et al. show examples for ISGs that are bound by the ISGF3 complex or by STAT2-IRF9 after IFN-beta treatment. Ly6e was identified as an ISG which bound STAT2-IRF9 (Sup. Dataset 1). Since authors of this manuscript use the Ly6e (Fig. 4E) as an example of STAT1/2 dependent gene regulation they should also comment on this finding.

We thank the reviewer for pointing us to this interesting aspect from a previous study (Platanitis et al, 2019). In our data we do not see a binding of STAT1 or STAT2 to the Ly6e promoter in ESCs or MEFs. It is noted that the promoter of Ly6e does not contain a canonical IRF motif. Rather we find that this locus gains STAT1/2 binding at adjacent putative enhancers that harbor multiple IRF binding sites. The differences might reflect cell type specific features and/or the ChIP-seq protocol used with respect to the anti-STAT2 antibodies or the method of chromatin fragmentation (Kidder et al, 2011). This is now mentioned in the discussion.

Minor comments:

38. Check Supplementary Table S2: total number ISGs in NPC all (244 instead of 204).

We thank the reviewer for spotting this typo and in **Supplementary Table S2** the correct number of 244 is now given.

Reviewer 3 (Significance (Required)):

39. The question addressed by the authors, how specific chromatin features affect STAT1 and STAT2 binding and ISG induction, is of high relevance for the field. However, the comparison of two different cell types as well as the special position of ESCs in the field not necessarily support the initial question. Further, many other cell types may be more important regarding regulation of ISG expression.

We appreciate that the reviewer shares our view on the high relevance of the study's research topic. To us, there are a variety of criteria to select cell types for identification of chromatin related features that affect STAT1 and STAT2 binding and the field will obviously profit from extending the type of work we have conducted to other cell types. However, in our view, the comparison between ESCs and MEFs is particularly interesting for chromatin mediated gene regulation as ESCs have a largely different chromatin state than differentiated cells (see response to point #26). Furthermore, the comparison of ESCs, NPCs and MEFs is a

particularly well-established model system for epigenetically regulated differences in gene expression that occur during differentiation and we have used it in a number of studies (Molitor et al, 2017; Teif et al, 2014; Teif et al, 2012; Thorn et al, 2022). We also like to point to the results presented in **Fig. 6**. They lead us to conclude that, despite their differences, the chromatin mediated regulatory principles for STAT binding derived from our study are preserved between ESCs and MEFs on the single nucleosome scale with respect to histone modifications and ATAC signal.

References

- Bae S, Lesch BJ (2020) H3K4me1 Distribution Predicts Transcription State and Poising at Promoters. *Front Cell Dev Biol* 8: 289
- Barski A, Cuddapah S, Cui K, Roh TY, Schones DE, Wang Z, Wei G, Chepelev I, Zhao K (2007) High-resolution profiling of histone methylations in the human genome. *Cell* 129: 823-837
- Bolivar S, Anfossi R, Humeres C, Vivar R, Boza P, Munoz C, Pardo-Jimenez V, Olivares-Silva F, Diaz-Araya G (2018) IFN-beta Plays Both Pro- and Anti-inflammatory Roles in the Rat Cardiac Fibroblast Through Differential STAT Protein Activation. *Front Pharmacol* 9: 1368
- Burke JD, Sonenberg N, Plataniias LC, Fish EN (2011) Antiviral effects of interferon-beta are enhanced in the absence of the translational suppressor 4E-BP1 in myocarditis induced by Coxsackievirus B3. *Antivir Ther* 16: 577-584
- Cheng J, Blum R, Bowman C, Hu D, Shilatifard A, Shen S, Dynlacht BD (2014) A role for H3K4 monomethylation in gene repression and partitioning of chromatin readers. *Mol Cell* 53: 979-992
- Edfors F, Danielsson F, Hallstrom BM, Kall L, Lundberg E, Ponten F, Forsstrom B, Uhlen M (2016) Gene-specific correlation of RNA and protein levels in human cells and tissues. *Mol Syst Biol* 12: 883
- Eggenberger J, Blanco-Melo D, Panis M, Brennand KJ, tenOever BR (2019) Type I interferon response impairs differentiation potential of pluripotent stem cells. *Proc Natl Acad Sci U S A* 116: 1384-1393
- Ernst J, Kheradpour P, Mikkelson TS, Shores N, Ward LD, Epstein CB, Zhang X, Wang L, Issner R, Coyne M, Ku M, Durham T, Kellis M, Bernstein BE (2011) Mapping and analysis of chromatin state dynamics in nine human cell types. *Nature* 473: 43-49
- Gordeeva O (2019) TGFbeta Family Signaling Pathways in Pluripotent and Teratocarcinoma Stem Cells' Fate Decisions: Balancing Between Self-Renewal, Differentiation, and Cancer. *Cells* 8: 1500
- Ivashkiv LB (2018) IFNgamma: signalling, epigenetics and roles in immunity, metabolism, disease and cancer immunotherapy. *Nat Rev Immunol* 18: 545-558
- Kidder BL, Hu G, Zhao K (2011) ChIP-Seq: technical considerations for obtaining high-quality data. *Nat Immunol* 12: 918-922
- Kok F, Rosenblatt M, Teusel M, Nizharadze T, Goncalves Magalhaes V, Dachert C, Maiwald T, Vlasov A, Wasch M, Tyufekchieva S, Hoffmann K, Damm G, Seehofer D, Boettler T, Binder M, Timmer J, Schilling M, Klingmuller U (2020) Disentangling molecular mechanisms regulating sensitization of interferon alpha signal transduction. *Mol Syst Biol* 16: e8955
- Kristensen DM, Kalisz M, Nielsen JH (2005) Cytokine signalling in embryonic stem cells. *APMIS* 113: 756-772

Lee CJ, An HJ, Cho ES, Kang HC, Lee JY, Lee HS, Cho YY (2020) Stat2 stability regulation: an intersection between immunity and carcinogenesis. *Exp Mol Med* 52: 1526-1536

Lim PSL, Meshorer E (2021) Organization of the Pluripotent Genome. *Cold Spring Harb Perspect Biol* 13: a040204

Liu X, Zhao J, Xue L, Zhao T, Ding W, Han Y, Ye H (2022) A comparison of transcriptome analysis methods with reference genome. *BMC Genomics* 23: 232

Maillard PV, Ciaudo C, Marchais A, Li Y, Jay F, Ding SW, Voinnet O (2013) Antiviral RNA interference in mammalian cells. *Science* 342: 235-238

Mallm JP, Iskar M, Ishaque N, Klett LC, Kugler SJ, Muino JM, Teif VB, Poos AM, Grossmann S, Erdel F, Tavernari D, Koser SD, Schumacher S, Brors B, Konig R, Remondini D, Vingron M, Stilgenbauer S, Lichter P, Zapatka M, Mertens D, Rippe K (2019) Linking aberrant chromatin features in chronic lymphocytic leukemia to transcription factor networks. *Mol Syst Biol* 15: e8339

Marinov GK, Kundaje A, Park PJ, Wold BJ (2014) Large-scale quality analysis of published ChIP-seq data. *G3 (Bethesda)* 4: 209-223

Marquis JF, Kapoustina O, Langlais D, Ruddy R, Dufour CR, Kim BH, MacMicking JD, Giguere V, Gros P (2011) Interferon regulatory factor 8 regulates pathways for antigen presentation in myeloid cells and during tuberculosis. *PLoS Genet* 7: e1002097

Molitor J, Mallm JP, Rippe K, Erdel F (2017) Retrieving Chromatin Patterns from Deep Sequencing Data Using Correlation Functions. *Biophys J* 112: 473-490

Platanitis E, Demiroz D, Schneller A, Fischer K, Capelle C, Hartl M, Gossenreiter T, Muller M, Novatchkova M, Decker T (2019) A molecular switch from STAT2-IRF9 to ISGF3 underlies interferon-induced gene transcription. *Nat Commun* 10: 2921

Poirier EZ, Buck MD, Chakravarty P, Carvalho J, Frederico B, Cardoso A, Healy L, Ulferts R, Beale R, Reis e Sousa C (2021) An isoform of Dicer protects mammalian stem cells against multiple RNA viruses. *Science* 373: 231-236

Teif VB, Beshnova DA, Vainshtein Y, Marth C, Mallm JP, Hofer T, Rippe K (2014) Nucleosome repositioning links DNA (de)methylation and differential CTCF binding during stem cell development. *Genome Res* 24: 1285-1295

Teif VB, Vainshtein Y, Caudron-Herger M, Mallm JP, Marth C, Hofer T, Rippe K (2012) Genome-wide nucleosome positioning during embryonic stem cell development. *Nat Struct Mol Biol* 19: 1185-1192

Thorn GJ, Clarkson CT, Rademacher A, Mamayusupova H, Schotta G, Rippe K, Teif VB (2022) DNA sequence-dependent formation of heterochromatin nanodomains. *Nat Commun* 13: 1861

Wang R, Wang J, Acharya D, Paul AM, Bai F, Huang F, Guo YL (2014) Antiviral responses in mouse embryonic stem cells: differential development of cellular mechanisms in type I interferon production and response. *J Biol Chem* 289: 25186-25198

December 13, 2022

RE: Life Science Alliance Manuscript #LSA-2022-01823

Dr. Karsten Rippe
German Cancer Research Center (DKFZ)
Genome Organization & Function
Im Neuenheimer Feld 280 - B066
Heidelberg 69120
Germany

Dear Dr. Rippe,

Thank you for submitting your revised manuscript entitled "Epigenetic signals that direct cell type specific interferon beta response in mouse cells". We would be happy to publish your paper in Life Science Alliance pending final revisions necessary to meet our formatting guidelines.

- please address Reviewer 1's final comments
- please consult our manuscript preparation guidelines <https://www.life-science-alliance.org/manuscript-prep> and make sure your manuscript sections are in the correct order
- please upload your manuscript text file as an editable doc file
- please upload both your main and supplementary figures as single files
- please add a Running Title, Summary Blurb, and a Category for your manuscript to our system
- please add the Twitter handle of your host institute/organization as well as your own or/and one of the authors in our system
- please add a Conflict of Interest statement to your main manuscript text
- please use the [10 author names, et al.] format in your references (i.e. limit the author names to the first 10)
- please add a separate section for your figure legends to the main manuscript text
- please upload your tables as separate editable doc or excel files or make sure they're included in the doc file of the manuscript
- the GEO accession should be made publicly accessible at this point
- please incorporate the Supplementary References into the main Reference list
- the Supplementary Data Sets can be listed with the other Figure Legends, along with their description. Supplementary Table 6 can therefore be removed.

A. FINAL FILES:

B. MANUSCRIPT ORGANIZATION AND FORMATTING:

Sincerely,

Reviewer #1 (Comments to the Authors (Required)):

The authors have improved the manuscript with modifications to text and figures which added clarity.

A few minor comments:

1. In Fig. 1, the NPC data were removed, but the legend wasn't modified accordingly.
2. Please disregard comment 7 of my original review as irrelevant.
3. Not that it matters, but in the original fig. 5A the STAT2 track looks like an overlay of STAT2 and the H3K4me1 track.
4. I think adding data as suggested would have been a further improvement, but I agree that the manuscript in its present form contains a wealth of information.

Reviewer #2 (Comments to the Authors (Required)):

The study investigates the linkage between transcriptional induction of ISGs and STAT1/2 DNA binding in response to IFN-beta. To determine the transcriptional response authors use bulk as well as single-cell-RNA sequencing. The authors identify most important histone H3 modification as well as sites of open chromatin by ATAC. Main conclusions: i. Cell type specific differences in ISG expression levels were observed upon IFN-beta stimulation between ESCs and MEFs/NPCs. This includes the set of genes induced and expression strength; ii. Binding of both factors, STAT1 and STAT2, at promoter sites correlates partly with ISG activation; iii. Authors identify a fraction of ISGs that lack a promoter bound STAT1/2 complex. Authors use chromatin co-accessibility analysis based on scATAC-sequencing to link induction of these genes with activity of distal enhancer elements; iv. Pre-existing active chromatin marks (H3K4me1 and H3K27ac) correlate with STAT1/2 binding while H3K27me3 modification impeded this interaction.

I reviewed the manuscript in March 2022 and acknowledge the revision of the manuscript by the authors. I like to attest, that the authors' responses to my comments are carefully crafted and that the revised version of the manuscript strongly supports their conclusions. I think the question addressed by the authors is of high relevance and their findings elucidate the link between ISG transcription, the chromatin landscape and DNA recruitment of STAT1/2.

Reviewer #3 (Comments to the Authors (Required)):

The authors have convincingly addressed the points I had raised.

We have addressed we the requested formatting issues as follows:

Reviewer 1's final comments have been addressed by correcting the legend to Figure 1 to clarify that the NPC gene expression data are given in Supplementary Fig. S1 and are no longer included in Fig. 1.

The manuscript has been formatted according to the journal guidelines:

- Manuscript text file has been uploaded an editable docx file
- Main and supplementary figures have been uploaded as single pdf files
- Running Title, Summary Blurb, and Category have been added to the manuscript
- The Twitter handles are @dkfz for institute and @KarstenRippe for the corresponding author and have been added to the online submission
- A Conflict of Interest statement was added to the main manuscript text
- References were formatted to journal style with a limit the author names to the first 10.
- Figure legends have been moved to a separate section at the end of the main manuscript text.
- The main manuscript text does not include tables. All supplementary tables are provided as xls files. All descriptions of table content are included in the corresponding xls files.
- The Supplementary Data Sets 1-3 are now referenced as Supplementary Tables S3-S5 and have been have integrated with the other Supplementary Tables in the order of referencing in the main text. All supplementary tables are now listed at the end of the main manuscript text and are provided as xls files.
- The sequencing data were deposited under GEO accession GSE160764 and are publicly accessible as of January 1, 2023.
- The Supplementary References have been incorporated into the main Reference list.
- Information on statistical tests and number of experiments for the reported data has been added to the figure legends.
- A table (in xls format) with the source data used to generate the figures has been added.
- The name of 2nd author Isabelle née Lander has been changed to Isabelle Seufert.

January 16, 2023

RE: Life Science Alliance Manuscript #LSA-2022-01823R

Dr. Karsten Rippe
German Cancer Research Center
Chromatin Networks
Im Neuenheimer Feld 280 - B066
Heidelberg 69120
Germany

Dear Dr. Rippe,

Thank you for submitting your Research Article entitled "Epigenetic signals that direct cell type specific interferon beta response in mouse cells". It is a pleasure to let you know that your manuscript is now accepted for publication in Life Science Alliance. Congratulations on this interesting work.

DISTRIBUTION OF MATERIALS:

Again, congratulations on a very nice paper. I hope you found the review process to be constructive and are pleased with how the manuscript was handled editorially. We look forward to future exciting submissions from your lab.

Sincerely,
